# Realistic Evaluation of Deep Partial-Label Learning Algorithms

**Wei Wang** [1,2]    **Dong-Dong Wu** [3]    **Jindong Wang** [4]    **Gang Niu** [2,3]
**Min-Ling Zhang** [3]    **Masashi Sugiyama** [2,1]

[1] The University of Tokyo, Chiba, Japan    [2] RIKEN, Tokyo, Japan
[3] Southeast University, Nanjing, China    [4] William & Mary, Williamsburg, VA, USA
`wangw@g.ecc.u-tokyo.ac.jp` `{dongdongwu1230,gang.niu.ml}@gmail.com`
`jwang80@wm.edu` `zhangml@seu.edu.cn` `sugi@k.u-tokyo.ac.jp`

## Abstract

Partial-label learning (PLL) is a weakly supervised learning problem in which each example is associated with multiple candidate labels and only one is the true label. In recent years, many deep PLL algorithms have been developed to improve model performance. However, we find that some early developed algorithms are often underestimated and can outperform many later algorithms with complicated designs. In this paper, we delve into the empirical perspective of PLL and identify several critical but previously overlooked issues. First, model selection for PLL is non-trivial, but has never been systematically studied. Second, the experimental settings are highly inconsistent, making it difficult to evaluate the effectiveness of the algorithms. Third, there is a lack of real-world image datasets that can be compatible with modern network architectures. Based on these findings, we propose PLENCH, the first Partial-Label learning bENCHmark to systematically compare state-of-the-art deep PLL algorithms. We investigate the model selection problem for PLL for the first time, and propose novel model selection criteria with theoretical guarantees. We also create Partial-Label CIFAR-10 (PLCIFAR10), an image dataset of human-annotated partial labels collected from Amazon Mechanical Turk, to provide a testbed for evaluating the performance of PLL algorithms in more realistic scenarios. Researchers can quickly and conveniently perform a comprehensive and fair evaluation and verify the effectiveness of newly developed algorithms based on PLENCH. We hope that PLENCH will facilitate standardized, fair, and practical evaluation of PLL algorithms in the future.[1]

## 1 Introduction

Partial-label learning (PLL) is a weakly supervised learning problem that has attracted much attention recently (Sugiyama et al., 2022; Wang et al., 2022a; Tian et al., 2023). In PLL, each training example is associated with multiple candidate labels (Jin & Ghahramani, 2002; Cour et al., 2011). The true label for each example is hidden in the set of candidate labels, but not accessible to the learning algorithm. PLL has been successfully applied to computer vision (Liu & Dietterich, 2012; Zeng et al., 2013; Chen et al., 2018; Gong et al., 2018; Tang et al., 2023; Wang et al., 2024c), natural language processing (Garrette & Baldridge, 2013; Zhou et al., 2018; Ren et al., 2016a;b), web mining (Luo & Orabona, 2010), ecoinformatics (Briggs et al., 2012; Wang et al., 2019; Li et al., 2021; Lyu et al., 2022), etc.

Among various strategies to address this problem, deep learning-based PLL algorithms have demonstrated satisfactory generalization performance due to the strong representation learning capabilities of deep neural networks (Lv et al., 2020; Wang et al., 2022b). Despite the abundance of algorithms in this area, we find that there are several fundamental and critical issues that have received less attention in the PLL literature, including neglected model selection issues, inconsistent experimental settings, and lack of real-world image datasets, which will be discussed point by point.

---

[1] The code implementation of PLENCH is available at `https://github.com/wwangwitsel/PLENCH`. The PLCIFAR10 dataset is available at `https://github.com/wwangwitsel/PLCIFAR10`.

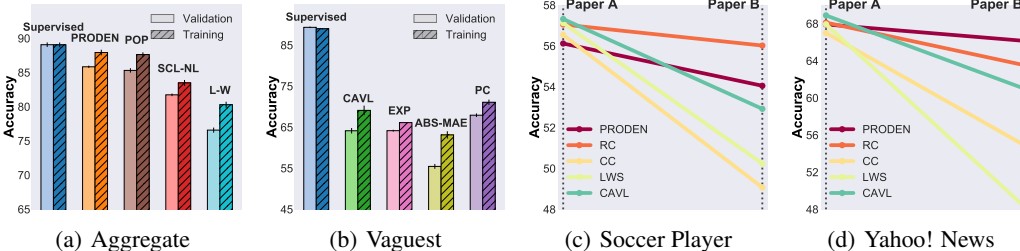

Figure 1: The two left panels show the differences in using an ordinary-label dataset for validation (lighter colors) and training (darker colors) for a given algorithm. For validation (lighter colors), we searched for the best hyperparameter configurations with the validation set for a given algorithm. For training (darker colors), we considered the validation set as partial-label examples with a single partial label and added them to the training set for training, using default hyperparameters without tuning. For fair comparisons, we trained all models with the same number of iterations. The two right panels show the classification accuracies of some PLL algorithms on Soccer Player and Yahoo! News from papers A (Zhang et al., 2022) and B (Xu et al., 2023b), respectively.

**Neglected model selection issues.** Most PLL algorithms select their hyperparameters by using a clean ordinary-label validation set (Qiao et al., 2023a; Xu et al., 2023a;b). However, the original definition of PLL does not allow the existence of an ordinary-label dataset (Jin & Ghahramani, 2002; Cour et al., 2011; Zhang et al., 2017), indicating a mismatch between the problem definitions and experimental settings in the literature. This problem can even lead to *unfair comparisons* if some algorithms follow the classical protocol of PLL to prohibit the use of ordinary-label data, while some algorithms do not. Moreover, *if we have a clean ordinary-label dataset, why not use it for training?* In many cases, the use of clean labels is more valuable for weakly supervised learning than for ordinary supervised learning (Hendrycks et al., 2018; Yu et al., 2023). A pilot experiment was conducted to compare different approaches to using validation data with ordinary labels. Figures 1(a) and 1(b) illustrate the results of the experiment on two versions of our collected PLCIFAR10 dataset. We can observe that for many PLL algorithms without much need to tune hyperparameters, it is more beneficial to use validation data with ordinary labels for training. Therefore, it is imperative to standardize the use of validation data and model selection criteria to ensure the integrity and practicality of empirical studies.

**Inconsistent experimental settings.** We have found that the experimental settings used in different papers are often quite different, creating a dilemma when comparing the performance of different algorithms. We illustrate this point with an example in Figures 1(c) and 1(d). We reported experimental results of several PLL algorithms on Soccer Player and Yahoo! News from papers A (Zhang et al., 2022) and B (Xu et al., 2023b), respectively. The data points for the same algorithm are connected by a line, and the number of intersections between different lines indicates the times of inconsistency in the performance ranking. In particular, the algorithm implementations, network architectures, and datasets are identical. However, the relative ranking of the classification performance differed significantly between the different papers. It is hypothesized that the discrepancy is due to subtle variations in experimental settings, such as data partitioning and preprocessing, as well as hyperparameter configurations. Such an obstacle can hinder objective comparisons of different algorithms, making it difficult to determine the effectiveness of a developed technique.

**Lack of real-world image datasets.** Existing PLL works have mainly conducted experimental results on real-world tabular datasets or synthetic image datasets to demonstrate the effectiveness of the proposed methodology (Lyu et al., 2021; Wang et al., 2022b). However, on the one hand, tabular datasets may not be compatible with modern network architectures, such as convolutional neural networks (Oquab et al., 2015; He et al., 2016). On the other hand, synthetic image datasets are generated by a human-made generation process (see Section 2), which may not be consistent with complex annotation mechanisms in real-world applications (Jiang et al., 2020; Wei et al., 2022). This may lead to potential concerns about whether an algorithm that works well on synthetic image datasets will still work well on real-world complex image datasets (Xu et al., 2021).

**Contributions.** To this end, we propose PLENCH, the first Partial-Label learning bENCHmark to standardize the experiments of PLL. The main contributions are as follows:

- We systematically investigate the model selection problem in PLL for the first time and propose new model selection criteria that are both theoretically and empirically validated.
- We create PLCIFAR10, a new PLL benchmark dataset with human-annotated partial labels. PLCIFAR10 provides an effective testbed for evaluating the performance of PLL algorithms in more realistic scenarios.
- We present the first PLL benchmark that includes twenty-seven algorithms and eleven real-world datasets, to systematically compare state-of-the-art deep PLL algorithms.

**Takeaways.** Based on our research, we suggest the following key takeaways:

- Many concise algorithms can outperform or be comparable to complicated algorithms with strong regularization techniques and higher computational requirements.
- There is no algorithm that can outperform all other algorithms in all cases, suggesting that the algorithmic design should be tailored for different cases.
- Model selection is non-trivial for PLL, and should be specified when proposing a PLL algorithm or comparing different algorithms.

## 2 BACKGROUND

**Problem setting.** Let $\mathcal{X} = \mathbb{R}^d$ denote the $d$-dimensional feature space and $\mathcal{Y} = \{1, 2, \ldots, q\}$ denote the label space with $q$ class labels. Let $(\boldsymbol{x}, S)$ denote a partial-label example where $\boldsymbol{x} \in \mathcal{X}$ is a feature vector and $S \subseteq \mathcal{Y}$ is a candidate label set associated with $\boldsymbol{x}$. The basic assumption of PLL is that the ground-truth label $y$ of $\boldsymbol{x}$ is concealed within its candidate label set $S$, i.e., $y \in S$. The task of PLL is to learn a multi-class classifier $\boldsymbol{f} : \mathcal{X} \rightarrow [0, 1]^q$ from a partial-label training set $\mathcal{D}^{\mathrm{Tr}} = \{(\boldsymbol{x}_i^{\mathrm{Tr}}, S_i^{\mathrm{Tr}})\}_{i=1}^{n^{\mathrm{Tr}}}$ where $\boldsymbol{f}(\boldsymbol{x}) = [f_1(\boldsymbol{x}), f_2(\boldsymbol{x}), \ldots, f_q(\boldsymbol{x})]$ is the estimated class-posterior probability vector for $\boldsymbol{x}$.

**Data generation process.** There are mainly two ways to generate synthetic *instance-independent* partial labels, i.e., the uniform sampling strategy (USS) (Feng et al., 2020b;a) and the flipping probability strategy (FPS) (Zhang et al., 2022). Feng et al. (2020b) proposed the USS that the candidate label set containing the ground-truth label is sampled from a uniform distribution, i.e.,

$$p(S|\boldsymbol{x}, y) = \frac{1}{2^{q-1} - 1} \mathbb{I}(y \in S), \tag{1}$$

where $\mathbb{I}(\pi)$ returns 1 if predicate $\pi$ holds and returns 0 otherwise. Wen et al. (2021) proposed the FPS that each false positive label is independently drawn into the candidate label set with a flipping probability $p(z \in S|y)$. Then, the class-conditional probability distribution of the candidate label set could be formulated as

$$p(S|\boldsymbol{x}, y) = \prod_{m \in S, m \neq y} p(m \in S|y) \cdot \prod_{t \notin S} (1 - p(t \in S|y)). \tag{2}$$

Although FPS seems more practical, the flipping probability is unknown. Many papers assume that the flipping probability is a constant value for different labels, which is often not true in real-world scenarios (Wei et al., 2022). Xu et al. (2021) proposed *instance-dependent* PLL, which assumes that the generation of partial labels also depends on the feature. However, the datasets are still synthesized by using the model outputs of an auxiliary network. In Section 4, we present a more realistic benchmark dataset with human-annotated partial labels.

## 3 REALISTIC MODEL SELECTION CRITERIA FOR PLL

Model selection is a critical component in machine learning problems. However, it has received little attention in the context of PLL. Most PLL work assumes the existence of a clean ordinary-label dataset, which may be neither consistent with the original definition of PLL nor practical, as discussed in Section 1. To promote fair and practical evaluation of PLL algorithms, it is important to systematically examine the model selection procedure for PLL. We follow the original definition of PLL to have only a single partial-label training set (Cour et al., 2011; Zhang et al., 2017). Then,

following a widely used validation procedure in machine learning (Raschka, 2018; Gulrajani & Lopez-Paz, 2021), we divide a partial-label validation set $\mathcal{D}^{\text{Val}} = \{(\boldsymbol{x}_i^{\text{Val}}, S_i^{\text{Val}})\}_{i=1}^{n^{\text{Val}}}$ from the training set for model selection. Next, we introduce each model selection criterion in turn.

### 3.1 COVERING RATE

**Definition 1** (Covering Rate (CR))**.** The Covering Rate of a multi-class classifier $\boldsymbol{f}$ on the partial-label validation set $\mathcal{D}^{\text{Val}}$ is defined as:

$$\text{CR}(\boldsymbol{f}) = \frac{1}{n^{\text{Val}}} \sum_{i=1}^{n^{\text{Val}}} \mathbb{I}\left(\arg\max_j f_j(\boldsymbol{x}_i^{\text{Val}}) \in S_i^{\text{Val}}\right). \tag{3}$$

CR indicates the fraction of validation data whose predicted label is included in its candidate label set. It is a natural metric in PLL, but its effectiveness may depend on the size of the candidate label sets. When the number of partial labels of each example is equal to 1, i.e., $|S| = 1$, PLL is reduced to ordinary supervised learning and CR is reduced to the validation accuracy. However, as $|S|$ increases, more false positive labels are included in the candidate label set. In the most extreme case, when $|S| = q$, PLL is reduced to unsupervised learning and CR does not convey effective information. Before analyzing the gap between CR and the validation accuracy, we introduce the following definition.

**Definition 2** (Ambiguity Degree)**.** The Ambiguity Degree $\gamma$ is defined as

$$\gamma = \sup_{(\boldsymbol{x},y)\sim p(\boldsymbol{x},y), S\sim p(S|\boldsymbol{x},y), \bar{y}\neq y} p(\bar{y} \in S), \tag{4}$$

where $p(\boldsymbol{x}, y)$ is the joint distribution over $\boldsymbol{x}$ and $y$.

If $\gamma < 1$, the small ambiguity degree is satisfied and the ERM learnability for PLL is guaranteed (Cour et al., 2011; Liu & Dietterich, 2014). We also define the expected accuracy as

$$\text{ACC}(\boldsymbol{f}) = \mathbb{E}_{p(\boldsymbol{x},y)}\mathbb{I}(\arg\max_l f_l(\boldsymbol{x}) = y). \tag{5}$$

Then, we have the following proposition.

**Proposition 1.** *Suppose that there is a constant $\epsilon \in (0,1)$ such that the expected accuracy of a classifier $\boldsymbol{f}$ satisfies $\text{ACC}(\boldsymbol{f}) \geqslant \epsilon$. Then, we have $\mathbb{E}[\text{CR}(\boldsymbol{f})] - \text{ACC}(\boldsymbol{f}) \leqslant (1-\epsilon)\gamma$.*

The proof is in Appendix A. The gap between CR and the accuracy is affected by both the accuracy and the ambiguity degree. If the classifier is more accurate and the partial-label dataset is less ambiguous, the gap between the CR metric and the accuracy will be smaller. Furthermore, we show that the minimizers of both are the same under certain conditions.

**Theorem 1.** *Suppose that the partial labels are generated by following the USS or the FPS with a constant flipping probability. Then, for any two classifiers $\boldsymbol{f}_1$ and $\boldsymbol{f}_2$ that satisfy $\mathbb{E}[\text{CR}(\boldsymbol{f}_1)] < \mathbb{E}[\text{CR}(\boldsymbol{f}_2)]$, we have $\text{ACC}(\boldsymbol{f}_1) < \text{ACC}(\boldsymbol{f}_2)$.*

The proof is in Appendix A. Theorem 1 shows that when partial labels are generated by using the USS or the FPS with a constant flipping probability, the classifier that minimizes the expectation of CR will also minimize the expected accuracy. Therefore, the CR metric will serve as a *consistent* model selection criterion for PLL under certain data distribution assumptions. However, this conclusion may not hold when partial labels are not generated by either strategy.

### 3.2 APPROXIMATED ACCURACY

Next, we introduce the definition of the Approximated Accuracy metric.

**Definition 3** (Approximated Accuracy (AA))**.** The Approximated Accuracy of a multi-class classifier $\boldsymbol{f}$ on the partial-label validation set $\mathcal{D}^{\text{Val}}$ is defined as:

$$\text{AA}(\boldsymbol{f}) = \frac{1}{n^{\text{Val}}} \sum_{i=1}^{n^{\text{Val}}} \sum_{j \in S_i^{\text{Val}}} \frac{f_j(\boldsymbol{x}_i^{\text{Val}})}{\sum_{k \in S_i^{\text{Val}}} f_k(\boldsymbol{x}_i^{\text{Val}})} \mathbb{I}\left(\arg\max_l f_l(\boldsymbol{x}_i^{\text{Val}}) = j\right). \tag{6}$$

Then, we have the following theorem.

**Theorem 2.** *Suppose there is a function $C : \mathcal{X} \times 2^{\mathcal{Y}} \mapsto \mathbb{R}$ such that the condition $p(S|\boldsymbol{x}, y) = C(\boldsymbol{x}, S)\mathbb{I}(y \in S)$ holds for partial-label data. Suppose further that the multi-class classifier $\boldsymbol{f}(\boldsymbol{x})$ is consistent with $p(y|\boldsymbol{x})$. Then, under mild conditions $\mathrm{AA}(\boldsymbol{f})$ is statistically consistent with the expected accuracy, i.e., $\mathbb{E}\left[\mathrm{AA}(\boldsymbol{f})\right] = \mathrm{ACC}\left(\boldsymbol{f}\right)$.*

The proof can be found in Appendix A. The introduction of AA is inspired by data-generation-based strategies for PLL (Wu et al., 2023). Theorem 2 illustrates that AA can be a consistent metric w.r.t. the expected classification accuracy on test data under certain assumptions. In particular, the data distribution assumption can hold for a wide range of types of partial labels (Liu & Dietterich, 2012; Wu et al., 2023). However, there are two factors that may affect its effectiveness. First, it may not be suitable for certain algorithms (Wen et al., 2021) where the loss function is not *strictly proper* (Gneiting & Raftery, 2007; Charoenphakdee et al., 2021) and its modeling output is not calibrated to the posterior probabilities. Second, it requires that the modeling output to be accurate, which may not be satisfied in the early stages of training.

### 3.3 ORACLE ACCURACY

Finally, we present the definition of the Oracle Accuracy metric.

**Definition 4** (Oracle Accuracy (OA)). The Oracle Accuracy of a multi-class classifier $\boldsymbol{f}$ on a partial-label validation set $\mathcal{D}^{\mathrm{Val}}$ is defined as

$$\mathrm{OA}(\boldsymbol{f}) = \frac{1}{n^{\mathrm{Val}}} \sum_{i=1}^{n^{\mathrm{Val}}} \mathbb{I}\left(\arg\max_{l} f_l(\boldsymbol{x}_i^{\mathrm{Val}}) = y_i^{\mathrm{Val}}\right), \qquad (7)$$

where $y_i^{\mathrm{Val}}$ is the underlying true label of $\boldsymbol{x}_i^{\mathrm{Val}}$.

The OA metric is natural if we have access to true labels in supervised learning. However, such a condition is not realistic in real problems of PLL. In fact, the availability of true labels contradicts the original motivation of PLL, which is to reduce labeling costs at the expense of true label ignorance. Inspired by Gulrajani & Lopez-Paz (2021), we restrict the use of true labels by allowing only one query (the last checkpoint) for each hyperparameter configuration. This means that we do not allow early stopping when using the OA metric. We try to compensate for the unrealistic access to true labels by restricting the model selection space. We also include the results of OA with early stopping (ES), which can be considered as an upper bound of the model selection performance, for reference. Actually, OA with ES may be the most common model selection criterion for PLL papers.

## 4 PLCIFAR10: A DATASET WITH HUMAN-ANNOTATED PARTIAL LABELS

In this section, we present a novel image dataset with human-annotated partial labels to compensate for the lack of real-world image datasets for PLL. We used Amazon Mechanical Turk (MTurk) as our crowdsourced annotation platform. CIFAR-10 (Krizhevsky & Hinton, 2009) was chosen as the base dataset because it has been widely used in the PLL literature. In addition, the difficulty and workload are relatively moderate, which can facilitate our experimental design and analysis. We posted the annotation tasks of the images from the training set of CIFAR-10 as Human Intelligence Tasks (HITs), and the crowdsourced workers received salaries by completing the HITs. We created 5000 HITs, each containing 10 random images. Workers were allowed to select *multiple candidate labels*, which could include the true label for an image. We then asked 10 different workers to perform the same HIT at the same time. As a result, each image could be annotated with multiple candidate label sets, and each candidate label set could contain multiple labels. More details on the data collection can be found in Appendix B.

In summary, we collected 502,190 candidate label sets with a total of 712,109 partial labels. Figure 2(a) shows the distribution of our collected partial labels. The imbalanced distribution of partial labels may not match the commonly used USS or FPS with a constant probability. Furthermore, we find that the noise rate, i.e., the proportion of training examples whose candidate label sets do not contain the true label, is high with few annotators, as shown in Figure 2(b). As the number of annotators increases, the aggregation of partial labels may become less noisy. We consider two

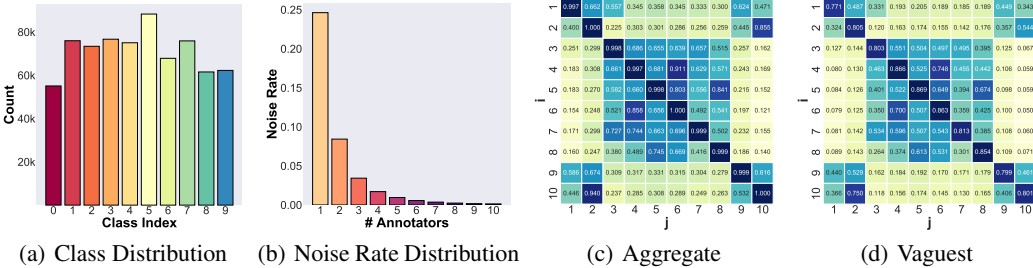

(a) Class Distribution    (b) Noise Rate Distribution    (c) Aggregate    (d) Vaguest

Figure 2: (a). The distribution of the collected partial labels of PLCIFAR10. (b) The noise rate with the increasing number of annotators. (c) The flipping probability matrix computed on PLCIFAR10-Aggregate. (d) The flipping probability matrix computed on PLCIFAR10-Vaguest.

versions of PLCIFAR10 for experiments. The first is **PLCIFAR10-Aggregate**, which assigns the aggregation of all partial labels from all annotators to each example. The second is **PLCIFAR10-Vaguest**, which assigns to each example the largest candidate label set from the annotators. Since PLCIFAR10-Vaguest has a high noise rate, it actually serves as a dataset for noisy PLL (Xu et al., 2023a; Lv et al., 2024a), which is a more practical setting since it is hard to ensure that annotated partial labels always include the true label in real-world applications. Figures 2(c) and 2(d) show the flipping probability matrices $p(j \in S|y = i)$ computed based on the two versions of PLCIFAR10. We can see that the flipping probabilities are not equal and the diagonals are not all 1, which is more challenging and practical than current synthetic datasets generated with the USS or the FPS.

## 5 SETUP OF PLENCH

### 5.1 BENCHMARK ALGORITHMS

We have divided the benchmark algorithms into four main groups. The detailed descriptions and hyperparameter settings can be found in Appendix E.

**Vanilla deep PLL algorithms.** We considered deep PLL algorithms that used simple loss functions or disambiguation strategies without strong regularization techniques as vanilla deep PLL algorithms. Identification-based strategies included PRODEN (Lv et al., 2020), CAVL (Zhang et al., 2022), and POP (Xu et al., 2023b). Note that RC (Feng et al., 2020b) is in the same form as PRODEN, so we only included PRODEN here to avoid repetitive comparisons. The averaging-based strategies included ABS-MAE (Lv et al., 2024a) and ABS-GCE (Lv et al., 2024a), which showed favorable performance among the losses in the family. The data-generation-based strategies included EXP (Feng et al., 2020a), MCL-GCE (Feng et al., 2020a), MCL-MSE (Feng et al., 2020a), CC (Feng et al., 2020b), LWS (Wen et al., 2021), and IDGP (Qiao et al., 2023a). Also, LOG (Feng et al., 2020a) is analogous to CC, so we only included CC here.

**Vanilla deep CLL algorithms.** Complementary-label learning (CLL) is a special case of PLL with only one label excluded from each candidate label set of training examples, i.e., $|S| = q - 1$ (Wang et al., 2024b). If we consider each label outside the candidate label set as a complementary label, it is possible to apply CLL algorithms to solve PLL problems as well (Feng et al., 2020a). Therefore, we included several vanilla CLL algorithms with simple loss functions in PLENCH. The vanilla CLL algorithms included PC (Ishida et al., 2017), Forward (Yu et al., 2018), NN (Ishida et al., 2019), GA (Ishida et al., 2019), SCL-EXP (Chou et al., 2020), SCL-NL (Chou et al., 2020), L-W (Gao & Zhang, 2021), and OP-W (Liu et al., 2023).

**Holistic deep PLL algorithms.** We considered PLL algorithms that employ strong representation learning or regularization techniques to be holisitic PLL algorithms. We used five algorithms, including VALEN (Xu et al., 2021), PiCO (Wang et al., 2022b), ABLE (Xia et al., 2022), CRDPLL (Wu et al., 2022), and DIRK (Wu et al., 2024).

**Deep noisy PLL algorithms.** We also included three noisy PLL algorithms that explicitly considered tailored strategies to handle the cases where the true label might be outside the candidate

label set. The deep noisy PLL algorithms included FREDIS (Qiao et al., 2023b), ALIM (Xu et al., 2023a), and PiCO+ (Wang et al., 2024a).

## 5.2 BENCHMARK DATASETS

To comprehensively evaluate the model performance of all algorithms, we included eleven real-world PLL benchmark datasets, including nine widely used tabular datasets and two image datasets that we collected. The tabular datasets included Lost (Cour et al., 2011), Soccer Player (Zeng et al., 2013), and Yahoo! News (Guillaumin et al., 2010) for the automatic face naming task, MSRCv2 (Liu & Dietterich, 2012) for the object classification task, Mirflickr (Huiskes & Lew, 2008) for the web image classification task, Birdsong (Briggs et al., 2012) for the bird song classification task, Malagasy (Garrette & Baldridge, 2013), Italian (Johan et al., 2009), and English (Zhou et al., 2018) for the POS tagging task. A detailed description of the datasets can be found in Appendix B.

## 5.3 EXPERIMENTAL SETTINGS

In this paper, we mainly investigate the hyperparameter tuning problem in the context of model selection. For tabular datasets, we first divided a test set from the entire dataset. Since the datasets were not explicitly divided into training and validation parts, we manually divided them into a partial-label training set $\mathcal{D}^{\text{Tr}}$ and a partial-label validation set $\mathcal{D}^{\text{Val}}$. Then, we trained a model with $\mathcal{D}^{\text{Tr}}$ and evaluated its validation performance on $\mathcal{D}^{\text{Val}}$ with the model selection criteria proposed in Section 3 as well as its test performance on the test set $\mathcal{D}^{\text{Te}}$ with ordinary labels. We then selected the checkpoint with the best validation performance and returned the corresponding test accuracy as the final result. We randomly selected a set of hyperparameter configurations from a given pool for a given data split and recorded the mean accuracy as well as the standard deviations for the best performance with different dataset splits. We used ResNet (He et al., 2016) and DenseNet (Huang et al., 2017) for image datasets and a multilayer perceptron (MLP) with a hidden layer width of 500 equipped with the ReLU (Nair & Hinton, 2010) activation function for tabular datasets.

## 6 EXPERIMENTS

### 6.1 EXPERIMENTAL RESULTS ON TABULAR DATASETS

Figure 3 shows the box plots of vanilla deep PLL and CLL algorithms on real tabular datasets, where NN and GA are not included in the figure because their performance was not satisfactory. Moreover, most holistic deep PLL algorithms are based on different data augmentation strategies and cannot be applied here. Detailed experimental results can be found in Appendix F. We can observe that AA does not work well in some cases where the classifier is not accurate enough. The CR metric is sometimes more effective and can serve as an alternative model selection criterion. Therefore, we may need to prepare different model selection criteria to effectively determine the hyperparameters in different cases. The classification performance of PRODEN is very strong in most cases, which clearly confirms its effectiveness. However, it still cannot outperform all other algorithms in all cases. In addition, some early vanilla CLL algorithms, such as Forward, SCL-EXP, and OP-W, can also achieve good performance and deserve to be considered when conducting comparative experiments for PLL.

### 6.2 EXPERIMENTAL RESULTS ON IMAGE DATASETS

Tables 1 and 2 report the experimental results on PLCIFAR10-Aggregate and PLCIFAR10-Vaguest with ResNet. Tables 3 and 4 report the experimental results on PLCIFAR10-Aggregate and PLCIFAR10-Vaguest with DenseNet. We do not include the results of VALEN because it requires a very large computational and memory budget. The following conclusions can be drawn. First, compared to the results with OA and ES, CR is very effective in most cases. The performance of AA decreases in some cases. We speculate that this is because the modeling results may not be reliable, and the results of AA may deviate from the true accuracy. Second, PRODEN and its variants are still strong in performance, but there is no algorithm that can outperform all other algorithms in all cases. Third, noisy partial labels have a significant impact on the model performance of most algorithms. Therefore, it is still promising to develop effective noisy PLL algorithms in more practical scenarios.

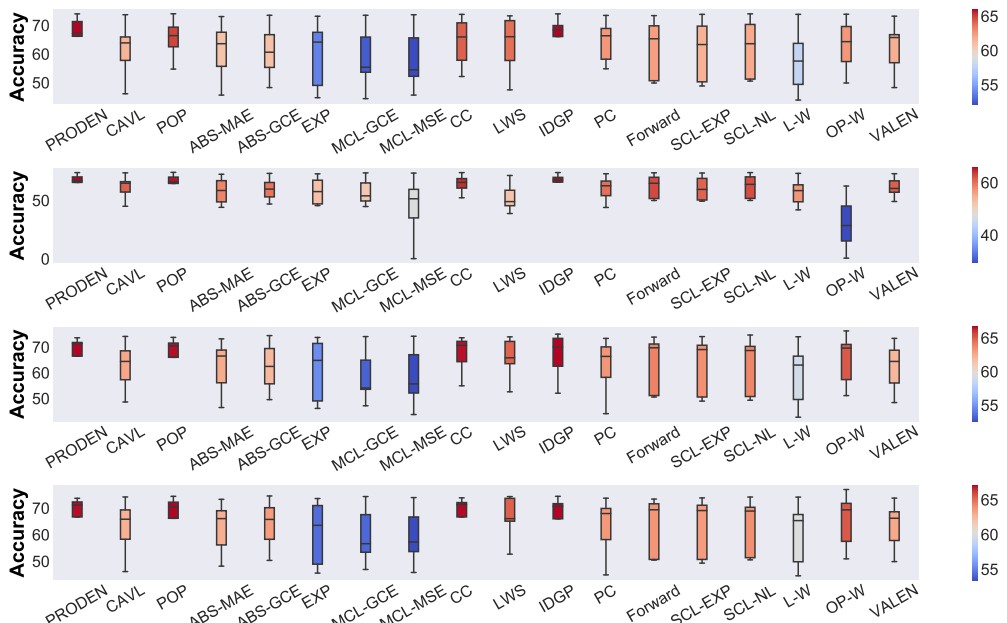

Figure 3: Experimental results of different algorithms on tabular datasets. The top, middle, and bottom figures correspond to box plots of experimental results using CR, AA, OA, and OA with ES for hyperparameter tuning, respectively. The colors of the bars indicate the mean accuracy.

Table 1: Classification accuracy (mean±std) of each algorithm on PLCIFAR10-Aggregate with ResNet, where the best performance w.r.t. each metric is shown in bold.

| Algorithm | Venue | w/ CR | w/ AA | w/ OA | w/ OA & ES |
|---|---|---|---|---|---|
| PRODEN | ICML 2020 (Lv et al., 2020) | **85.95±0.08** | 85.77±0.11 | 85.91±0.18 | 86.03±0.13 |
| CAVL | ICLR 2022 (Zhang et al., 2022) | 68.09±0.25 | 65.92±1.15 | 68.23±0.17 | 69.12±0.47 |
| POP | ICML 2023 (Xu et al., 2023b) | 84.94±0.08 | 85.27±0.34 | 85.04±0.27 | 85.53±0.24 |
| ABS-MAE | TPAMI 2024 (Lv et al., 2024a) | 55.13±1.13 | 45.51±4.38 | 54.31±1.22 | 55.32±1.15 |
| ABS-GCE | TPAMI 2024 (Lv et al., 2024a) | 74.21±1.08 | 71.28±1.20 | 74.68±0.80 | 76.00±0.44 |
| EXP | ICML 2020 (Feng et al., 2020a) | 10.00±0.00 | 10.00±0.00 | 10.00±0.00 | 10.00±0.00 |
| MCL-GCE | ICML 2020 (Feng et al., 2020a) | 60.57±0.60 | 58.26±1.14 | 32.31±9.73 | 62.26±0.72 |
| MCL-MSE | ICML 2020 (Feng et al., 2020a) | 61.03±1.25 | 10.00±0.00 | 55.09±1.96 | 63.17±0.59 |
| CC | NeurIPS 2020 (Feng et al., 2020b) | 80.66±0.23 | 80.77±0.46 | 81.44±0.29 | 81.87±0.11 |
| LWS | ICML 2021 (Wen et al., 2021) | 55.31±0.32 | 26.44±5.35 | 55.50±0.58 | 56.17±0.51 |
| IDGP | ICLR 2023 (Qiao et al., 2023a) | 82.80±0.30 | 80.72±0.66 | 83.43±0.17 | 83.65±0.44 |
| PC | NeurIPS 2017 (Ishida et al., 2017) | 71.45±0.71 | 70.34±0.39 | 71.06±0.56 | 72.24±0.75 |
| Forward | ECCV 2018 (Yu et al., 2018) | 81.19±0.24 | 79.51±0.33 | 80.98±0.49 | 81.57±0.38 |
| NN | ICML 2019 (Ishida et al., 2019) | 30.68±0.39 | 25.97±1.17 | 29.50±0.51 | 31.65±0.38 |
| GA | ICML 2019 (Ishida et al., 2019) | 37.81±1.00 | 37.51±1.19 | 36.84±1.37 | 38.26±0.79 |
| SCL-EXP | ICML 2020 (Chou et al., 2020) | 79.50±0.33 | 79.57±0.37 | 79.34±0.53 | 80.30±0.15 |
| SCL-NL | ICML 2020 (Chou et al., 2020) | 81.87±0.07 | 81.09±0.55 | 79.96±0.56 | 81.75±0.14 |
| L-W | ICML 2021 (Gao & Zhang, 2021) | 76.76±0.49 | 74.04±0.87 | 74.76±0.60 | 76.76±0.49 |
| OP-W | AISTATS 2023 (Liu et al., 2023) | 78.91±0.22 | 78.71±0.39 | 79.64±0.21 | 81.15±0.49 |
| PiCO | ICLR 2022 (Wang et al., 2022b) | 79.20±0.61 | 75.19±0.59 | 79.37±0.50 | 79.88±0.77 |
| ABLE | IJCAI 2022 (Xia et al., 2022) | 85.86±0.18 | **86.30±0.14** | **86.07±0.08** | **86.46±0.10** |
| CRDPLL | ICML 2022 (Wu et al., 2022) | 81.60±0.55 | 79.82±0.16 | 81.66±0.52 | 82.36±0.19 |
| DIRK | AAAI 2024 (Wu et al., 2024) | 85.90±0.31 | 84.96±0.63 | 85.74±0.42 | 85.60±0.28 |
| FREDIS | ICML 2023 (Qiao et al., 2023b) | 85.01±0.08 | 84.94±0.07 | 85.66±0.22 | 85.74±0.09 |
| ALIM | NeurIPS 2023 (Xu et al., 2023a) | 63.51±0.73 | 61.76±1.57 | 61.42±1.16 | 64.72±0.27 |
| PiCO+ | TPAMI 2024 (Wang et al., 2024a) | 60.07±0.13 | 57.63±0.84 | 59.01±1.20 | 62.66±0.33 |

## 6.3 COMPLEXITY ANALYSIS

Figure 4 shows the running time and GPU memory usage for each running step of different PLL algorithms on PLCIFAR10-Vaguest with DenseNet. We can see that some holistic deep PLL al-

Table 2: Classification accuracy (mean±std) of each algorithm on PLCIFAR10-Vaguest with ResNet, where the best performance w.r.t. each metric is shown in bold.

| Algorithm | Venue | w/ CR | w/ AA | w/ OA | w/ OA & ES |
|---|---|---|---|---|---|
| PRODEN | ICML 2020 (Lv et al., 2020) | 74.95±0.09 | 68.54±0.39 | 74.78±0.64 | 76.94±0.41 |
| CAVL | ICLR 2022 (Zhang et al., 2022) | 63.62±0.27 | 61.76±0.75 | 63.70±0.62 | 64.68±0.18 |
| POP | ICML 2023 (Xu et al., 2023b) | 75.17±0.53 | 67.71±0.54 | 74.35±0.25 | 76.08±0.18 |
| ABS-MAE | TPAMI 2024 (Lv et al., 2024a) | 55.58±0.53 | 32.25±7.18 | 55.60±0.66 | 55.84±0.50 |
| ABS-GCE | TPAMI 2024 (Lv et al., 2024a) | 75.17±0.34 | 72.86±0.17 | 74.13±0.68 | 75.95±0.35 |
| EXP | ICML 2020 (Feng et al., 2020a) | 63.93±0.39 | 61.77±0.16 | 63.04±0.58 | 64.14±0.36 |
| MCL-GCE | ICML 2020 (Feng et al., 2020a) | 71.74±0.21 | 64.53±1.67 | 70.78±0.32 | 73.28±0.19 |
| MCL-MSE | ICML 2020 (Feng et al., 2020a) | 67.99±0.96 | 65.91±1.44 | 64.94±0.27 | 69.98±0.49 |
| CC | NeurIPS 2020 (Feng et al., 2020b) | 71.78±0.50 | 61.05±0.22 | 70.11±0.43 | 73.26±0.51 |
| LWS | ICML 2021 (Wen et al., 2021) | 60.21±0.59 | 44.12±10.76 | 60.98±0.46 | 61.78±0.67 |
| IDGP | ICLR 2023 (Qiao et al., 2023a) | 76.14±0.18 | 71.36±2.72 | 76.05±0.31 | 77.74±0.19 |
| PC | NeurIPS 2017 (Ishida et al., 2017) | 64.46±1.88 | 66.02±0.48 | 66.59±0.34 | 68.13±0.29 |
| Forward | ECCV 2018 (Yu et al., 2018) | 70.01±0.43 | 43.09±12.96 | 70.52±0.07 | 70.98±0.23 |
| NN | ICML 2019 (Ishida et al., 2019) | 33.23±0.32 | 25.31±1.05 | 30.35±0.37 | 33.12±0.48 |
| GA | ICML 2019 (Ishida et al., 2019) | 33.94±1.01 | 33.64±1.20 | 31.35±0.86 | 34.00±0.98 |
| SCL-EXP | ICML 2020 (Chou et al., 2020) | 71.13±0.51 | 65.45±0.58 | 71.33±0.50 | 72.86±0.28 |
| SCL-NL | ICML 2020 (Chou et al., 2020) | 69.66±0.50 | 60.46±1.54 | 69.56±0.47 | 71.52±0.14 |
| L-W | ICML 2021 (Gao & Zhang, 2021) | 69.81±0.59 | 59.19±1.45 | 71.12±0.19 | 72.44±0.45 |
| OP-W | AISTATS 2023 (Liu et al., 2023) | 70.53±0.51 | 69.64±2.67 | 73.20±0.26 | 73.21±0.10 |
| PiCO | ICLR 2022 (Wang et al., 2022b) | 74.06±0.22 | 68.33±0.58 | 73.27±0.46 | 74.70±0.33 |
| ABLE | IJCAI 2022 (Xia et al., 2022) | 75.49±0.58 | 68.14±0.25 | 74.87±0.49 | 76.27±0.13 |
| CRDPLL | ICML 2022 (Wu et al., 2022) | 76.21±0.58 | 70.99±0.19 | 75.70±0.09 | 77.68±0.46 |
| DIRK | AAAI 2024 (Wu et al., 2024) | **80.32±0.15** | **75.02±2.36** | **80.10±0.33** | **81.08±0.32** |
| FREDIS | ICML 2023 (Qiao et al., 2023b) | 74.57±0.90 | 67.72±0.13 | 73.45±0.32 | 76.94±0.24 |
| ALIM | NeurIPS 2023 (Xu et al., 2023a) | 65.49±1.05 | 66.51±0.83 | 67.57±1.93 | 70.59±1.21 |
| PiCO+ | TPAMI 2024 (Wang et al., 2024a) | 61.59±1.55 | 59.65±0.84 | 61.27±0.12 | 64.75±0.30 |

Table 3: Classification accuracy (mean±std) of each algorithm on PLCIFAR10-Aggregate with DenseNet, where the best performance w.r.t. each metric is shown in bold.

| Algorithm | Venue | w/ CR | w/ AA | w/ OA | w/ OA & ES |
|---|---|---|---|---|---|
| PRODEN | ICML 2020 (Lv et al., 2020) | 80.94±0.54 | **81.54±0.72** | 80.94±0.90 | 81.30±0.74 |
| CAVL | ICLR 2022 (Zhang et al., 2022) | 61.12±2.71 | 60.17±2.34 | 63.52±1.80 | 63.82±1.68 |
| POP | ICML 2023 (Xu et al., 2023b) | 80.99±0.31 | 80.90±0.53 | 81.20±0.41 | 81.38±0.43 |
| ABS-MAE | TPAMI 2024 (Lv et al., 2024a) | 53.93±2.12 | 50.56±1.85 | 52.56±1.68 | 53.85±2.05 |
| ABS-GCE | TPAMI 2024 (Lv et al., 2024a) | 72.48±0.34 | 71.50±0.46 | 73.05±0.71 | 74.28±0.70 |
| EXP | ICML 2020 (Feng et al., 2020a) | 10.00±0.00 | 10.00±0.00 | 10.00±0.00 | 10.00±0.00 |
| MCL-GCE | ICML 2020 (Feng et al., 2020a) | 58.82±0.31 | 57.81±0.24 | 36.47±10.98 | 61.34±0.03 |
| MCL-MSE | ICML 2020 (Feng et al., 2020a) | 57.29±0.22 | 10.00±0.00 | 54.27±2.35 | 59.37±0.38 |
| CC | NeurIPS 2020 (Feng et al., 2020b) | 78.28±0.82 | 77.64±0.67 | 78.52±0.38 | 78.97±0.49 |
| LWS | ICML 2021 (Wen et al., 2021) | 47.57±0.20 | 41.64±1.91 | 48.48±0.42 | 48.90±0.41 |
| IDGP | ICLR 2023 (Qiao et al., 2023a) | 77.49±0.86 | 76.07±0.80 | 78.41±0.84 | 79.03±0.86 |
| PC | NeurIPS 2017 (Ishida et al., 2017) | 65.60±0.13 | 65.95±0.54 | 66.12±0.34 | 66.55±0.47 |
| Forward | ECCV 2018 (Yu et al., 2018) | 78.74±0.48 | 78.02±0.30 | 78.38±0.39 | 79.39±0.19 |
| NN | ICML 2019 (Ishida et al., 2019) | 31.51±0.98 | 26.01±1.37 | 30.22±0.48 | 32.97±0.79 |
| GA | ICML 2019 (Ishida et al., 2019) | 37.21±0.58 | 36.85±0.46 | 37.28±0.18 | 37.86±0.20 |
| SCL-EXP | ICML 2020 (Chou et al., 2020) | 78.67±0.71 | 78.27±0.69 | 78.38±0.34 | 78.83±0.32 |
| SCL-NL | ICML 2020 (Chou et al., 2020) | 79.26±0.59 | 78.40±0.98 | 78.29±0.27 | 79.26±0.59 |
| L-W | ICML 2021 (Gao & Zhang, 2021) | 71.68±0.53 | 71.77±0.29 | 72.24±0.69 | 73.78±0.56 |
| OP-W | AISTATS 2023 (Liu et al., 2023) | 79.95±0.36 | 78.86±0.43 | 80.04±0.16 | 80.36±0.32 |
| PiCO | ICLR 2022 (Wang et al., 2022b) | 74.89±2.14 | 74.23±0.65 | 75.81±1.58 | 76.37±1.61 |
| ABLE | IJCAI 2022 (Xia et al., 2022) | **81.38±0.33** | 81.40±0.34 | **81.21±0.49** | 81.28±0.61 |
| CRDPLL | ICML 2022 (Wu et al., 2022) | 74.97±0.99 | 74.90±0.52 | 75.36±0.59 | 75.67±0.66 |
| DIRK | AAAI 2024 (Wu et al., 2024) | 77.74±0.64 | 77.83±0.53 | 77.86±0.67 | 77.85±0.76 |
| FREDIS | ICML 2023 (Qiao et al., 2023b) | 81.25±0.56 | 81.08±0.80 | 81.11±0.69 | **81.66±0.51** |
| ALIM | NeurIPS 2023 (Xu et al., 2023a) | 56.61±1.02 | 57.29±0.32 | 58.46±0.38 | 59.75±0.64 |
| PiCO+ | TPAMI 2024 (Wang et al., 2024a) | 57.05±0.71 | 54.01±1.71 | 56.02±1.02 | 58.45±0.15 |

gorithms with strong regularization terms and complicated training strategies can achieve better performance, but they also take more time and use more memory. Therefore, it is important to con-

Table 4: Classification accuracy (mean±std) of each algorithm on PLCIFAR10-Vaguest with DenseNet, where the best performance w.r.t. each metric is shown in bold.

| Algorithm | Venue | w/ CR | w/ AA | w/ OA | w/ OA & ES |
|---|---|---|---|---|---|
| PRODEN | ICML 2020 (Lv et al., 2020) | 72.73±0.61 | 67.42±1.06 | 72.53±0.48 | 73.98±0.11 |
| CAVL | ICLR 2022 (Zhang et al., 2022) | 55.93±1.95 | 58.85±1.00 | 59.01±1.29 | 59.81±1.49 |
| POP | ICML 2023 (Xu et al., 2023b) | 71.92±0.87 | 69.18±1.64 | 71.90±0.13 | 72.85±0.21 |
| ABS-MAE | TPAMI 2024 (Lv et al., 2024a) | 57.07±2.16 | 45.50±5.21 | 57.69±1.98 | 57.94±2.01 |
| ABS-GCE | TPAMI 2024 (Lv et al., 2024a) | **73.30±0.46** | 69.85±1.27 | 72.75±0.48 | 73.63±0.42 |
| EXP | ICML 2020 (Feng et al., 2020a) | 61.50±0.69 | 55.26±2.27 | 61.44±0.54 | 62.56±0.52 |
| MCL-GCE | ICML 2020 (Feng et al., 2020a) | 66.57±1.37 | 61.90±1.36 | 67.88±1.10 | 69.57±0.70 |
| MCL-MSE | ICML 2020 (Feng et al., 2020a) | 63.37±1.09 | 63.79±0.92 | 63.08±0.42 | 65.26±0.55 |
| CC | NeurIPS 2020 (Feng et al., 2020b) | 68.77±0.29 | 60.92±0.05 | 69.62±0.33 | 71.49±0.48 |
| LWS | ICML 2021 (Wen et al., 2021) | 57.85±1.47 | 54.65±1.65 | 58.16±2.38 | 59.09±2.00 |
| IDGP | ICLR 2023 (Qiao et al., 2023a) | 72.02±0.64 | 70.55±0.91 | **72.98±0.33** | **74.19±0.41** |
| PC | NeurIPS 2017 (Ishida et al., 2017) | 59.99±0.93 | 60.28±0.28 | 63.11±0.48 | 63.93±0.90 |
| Forward | ECCV 2018 (Yu et al., 2018) | 66.35±0.97 | 60.12±1.52 | 66.54±0.58 | 67.99±0.14 |
| NN | ICML 2019 (Ishida et al., 2019) | 31.08±0.91 | 26.52±0.37 | 30.23±0.63 | 33.54±0.12 |
| GA | ICML 2019 (Ishida et al., 2019) | 35.55±0.25 | 35.52±0.46 | 34.52±0.55 | 35.75±0.30 |
| SCL-EXP | ICML 2020 (Chou et al., 2020) | 68.99±0.34 | 65.59±0.82 | 69.41±0.84 | 70.72±0.49 |
| SCL-NL | ICML 2020 (Chou et al., 2020) | 66.90±0.37 | 61.42±0.22 | 65.87±1.03 | 68.71±0.64 |
| L-W | ICML 2021 (Gao & Zhang, 2021) | 68.28±0.83 | 63.52±0.71 | 68.20±0.48 | 69.66±0.43 |
| OP-W | AISTATS 2023 (Liu et al., 2023) | 69.91±1.42 | 70.25±0.84 | 71.55±0.49 | 72.55±0.41 |
| PiCO | ICLR 2022 (Wang et al., 2022b) | 71.76±0.28 | 69.30±0.93 | 70.89±0.59 | 72.27±0.48 |
| ABLE | IJCAI 2022 (Xia et al., 2022) | 71.68±1.01 | 68.22±1.08 | 72.10±0.20 | 73.44±0.51 |
| CRDPLL | ICML 2022 (Wu et al., 2022) | 70.73±0.75 | 70.69±0.17 | 71.99±0.67 | 72.46±0.29 |
| DIRK | AAAI 2024 (Wu et al., 2024) | 70.47±0.32 | **71.06±0.48** | 71.26±0.19 | 71.60±0.75 |
| FREDIS | ICML 2023 (Qiao et al., 2023b) | 71.38±0.41 | 65.92±0.13 | 71.33±0.30 | 72.95±0.25 |
| ALIM | NeurIPS 2023 (Xu et al., 2023a) | 61.91±1.23 | 62.04±1.10 | 63.84±0.49 | 65.37±0.72 |
| PiCO+ | TPAMI 2024 (Wang et al., 2024a) | 62.45±0.74 | 59.59±1.79 | 60.67±0.89 | 62.82±1.10 |

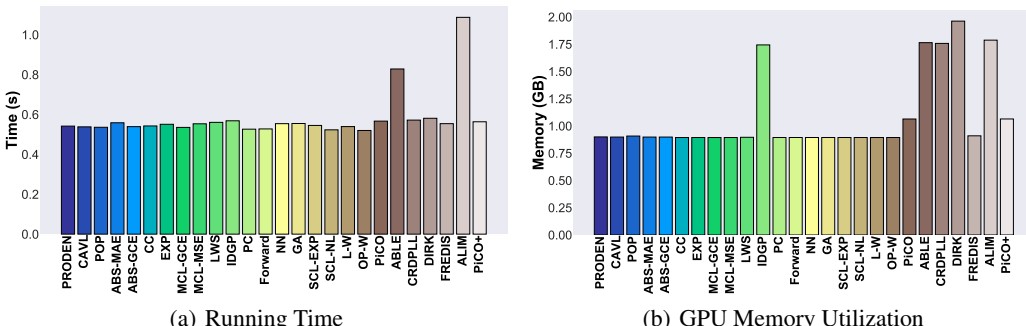

(a) Running Time        (b) GPU Memory Utilization

Figure 4: Running time and GPU memory utilization for each running step of different PLL algorithms on PLCIFAR10-Vaguest with DenseNet.

sider the tradeoff between performance and resource consumption when deciding which algorithm to use given limited computational resources.

## 7 CONCLUSION

In this paper, we proposed the first PLL benchmark to standardize the performance evaluation of state-of-the-art deep PLL algorithms. We proposed new model selection criteria to fill in the gap of model selection problems for PLL. We also introduced PLCIFAR10, a novel image dataset of human-annotated partial labels. We hope that the availability of this benchmark can promote fair, practical, and standardized evaluation of PLL algorithms in the future. A limitation of our work is that the datasets are relatively small, and we will leave the exploration of large-scale PLL datasets as our future work. In addition, our experiments are conducted with simple deep neural networks. It is also interesting to explore the incorporation of foundation models to improve the model performance of PLL algorithms in the future.

ACKNOWLEDGMENTS

The authors thank Wei-I Lin, Takashi Ishida, and Yu-Jie Zhang for their helpful discussions and suggestions, Wei-Xuan Bao for his help with the datasets, and Yuko Kawashima for her help with the funding management. The authors thank the anonymous reviewers for their helpful comments. WW was supported by the SGU MEXT Scholarship, by the Junior Research Associate (JRA) program of RIKEN, and by Microsoft Research Asia. MLZ was supported by National Science Foundation of China (62225602). MS was supported by the Institute for AI and Beyond, UTokyo and by a grant from Apple, Inc. Any views, opinions, findings, and conclusions or recommendations expressed in this material are those of the authors and should not be interpreted as reflecting the views, policies or position, either expressed or implied, of Apple Inc.

ETHICS STATEMENTS

This paper does not raise any ethical concerns. This paper contains a human subjects study where the data collection process was conducted on Amazon MTurk. The privacy of the annotators was strictly protected by Amazon MTurk. We strictly followed the Code of Ethics during the dataset collection process.

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

## A  PROOFS

### A.1  PROOF OF PROPOSITION 1

$$
\begin{aligned}
&\mathbb{E}\left[\mathrm{CR}(\boldsymbol{f})\right] - \mathrm{ACC}\left(\boldsymbol{f}\right)\\
&= \mathbb{E}_{p(\boldsymbol{x},S)}\left[\mathbb{I}\left(\arg\max_{j}\ f_j(\boldsymbol{x}) \in S\right)\right] - \mathbb{E}_{p(\boldsymbol{x},y)}\left[\mathbb{I}\left(\arg\max_{j}\ f_j(\boldsymbol{x}) = y\right)\right]\\
&= \mathbb{E}_{p(\boldsymbol{x},y,S)}\left[\mathbb{I}\left(\arg\max_{j}\ f_j(\boldsymbol{x}) \in S\backslash\{y\}\right)\right]\\
&= \mathbb{E}_{p(\boldsymbol{x},y,S)}\left[\mathbb{I}\left(\arg\max_{j}\ f_j(\boldsymbol{x}) \in S\backslash\{y\}\right)\mathbb{I}\left(\arg\max_{j}\ f_j(\boldsymbol{x}) \neq y\right)\right]\\
&= p\left(\arg\max_{j}\ f_j(\boldsymbol{x}) \in S\backslash\{y\},\ \arg\max_{j}\ f_j(\boldsymbol{x}) \neq y\right)\\
&= p\left(\arg\max_{j}\ f_j(\boldsymbol{x}) \neq y\right) p\left(\arg\max_{j}\ f_j(\boldsymbol{x}) \in S\backslash\{y\}|\arg\max_{j}\ f_j(\boldsymbol{x}) \neq y\right)\\
&= \left(1 - \mathrm{ACC}\left(\boldsymbol{f}\right)\right) p\left(\arg\max_{j}\ f_j(\boldsymbol{x}) \in S\backslash\{y\}|\arg\max_{j}\ f_j(\boldsymbol{x}) \neq y\right)\\
&\leqslant (1 - \epsilon)\max_{\bar{y}\neq y}\ p(\bar{y} \in S|\boldsymbol{x},y)\\
&\leqslant (1 - \epsilon)\gamma.
\end{aligned}
$$

Here, the third equation can be obtained by traversing the cases of $\arg\max_{j}\ f_j(\boldsymbol{x}) = y$, $\arg\max_{j}\ f_j(\boldsymbol{x}) \in S\backslash\{y\}$, and $\arg\max_{j}\ f_j(\boldsymbol{x}) \notin S$. The last inequality results from the definition of the ambiguity degree. The proof is completed. □

### A.2  PROOF OF THEOREM 1

Under the USS assumption or the FPS assumption with a constant flipping probability, the probability $p(\bar{y} \in S|\boldsymbol{x},\bar{y} \neq y)$ is a constant value for different $\bar{y}$. Suppose the constant is $c$. Then, we have

$$
\begin{aligned}
&\mathbb{E}\left[\mathrm{CR}(\boldsymbol{f})\right] - \mathrm{ACC}\left(\boldsymbol{f}\right)\\
&= \left(1 - \mathrm{ACC}\left(\boldsymbol{f}\right)\right) p\left(\arg\max_{j}\ f_j(\boldsymbol{x}) \in S\backslash\{y\}|\arg\max_{j}\ f_j(\boldsymbol{x}) \neq y\right)\\
&= c\left(1 - \mathrm{ACC}\left(\boldsymbol{f}\right)\right).
\end{aligned}
$$

Therefore, we have

$$
\mathrm{ACC}\left(\boldsymbol{f}\right) = \frac{\mathbb{E}\left[\mathrm{CR}(\boldsymbol{f})\right] - c}{1 - c}. \tag{8}
$$

Since we have $c < 1$, for any two classifiers $\boldsymbol{f}_1$ and $\boldsymbol{f}_2$ that satisfy $\mathbb{E}\left[\mathrm{CR}(\boldsymbol{f}_1)\right] < \mathbb{E}\left[\mathrm{CR}(\boldsymbol{f}_2)\right]$, we have $\mathrm{ACC}\left(\boldsymbol{f}_1\right) < \mathrm{ACC}\left(\boldsymbol{f}_2\right)$. The proof is completed. □

### A.3  PROOF OF THEOREM 2

The proof of Theorem 2 is mainly based on the theoretical results from Wu et al. (2023). We introduce the following lemma.

**Lemma 1** (Wu et al. (2023)). *Assume that there exist a function $C : \mathcal{X} \times 2^{\mathcal{Y}} \mapsto \mathbb{R}$ such that the condition $p(S|\boldsymbol{x},y) = C(\boldsymbol{x},S)\mathbb{I}\left(y \in S\right)$ holds for partial-label data. Then, the classification risk*

$$
\mathbb{E}_{p(\boldsymbol{x},y)}\left[\mathcal{L}(\boldsymbol{f}(\boldsymbol{x}),y)\right] \tag{9}
$$

*is equivalent to*

$$
\mathbb{E}_{p(\boldsymbol{x},y)}\left[\sum_{j\in S}\frac{p(y = j|\boldsymbol{x})}{\sum_{k\in S}p(y = k|\boldsymbol{x})}\mathcal{L}(\boldsymbol{f}(\boldsymbol{x}),k)\right]. \tag{10}
$$

Then, we provide the proof of Theorem 2.

*Proof of Theorem 2.* We set $\mathcal{L}(\boldsymbol{f}(\boldsymbol{x}), k) = \mathbb{I}\left(\arg\max_j \; f_j(\boldsymbol{x}) = y\right)$. Then, if the multi-class classifier $\boldsymbol{f}(\boldsymbol{x})$ is consistent with $p(y|\boldsymbol{x})$, $\mathrm{AA}(\boldsymbol{f})$ is statistically consistent with the expected accuracy. The proof is completed. □

## B  MORE DETAILS OF BENCHMARK DATASETS

In this section, we first describe the data sheet of PLCIFAR10, a novel dataset collected by us. Then, we provide more information of real-world datasets used in the paper. The summary of all the datasets is shown in Table 5.

Table 5: Characteristics of real-world PLL datasets used in PLENCH.

| Dataset | # Examples | # Features | # Classes | Avg. # CLs | Noise Rate | Type | Task Domain |
|---|---|---|---|---|---|---|---|
| Lost | 1,122 | 108 | 16 | 2.23 | 0% | tabular | automatic face naming (Cour et al., 2011) |
| MSRCv2 | 1,758 | 48 | 23 | 3.16 | 0% | tabular | object classification (Liu & Dietterich, 2012) |
| Mirflickr | 2,780 | 1,536 | 14 | 2.76 | 0% | tabular | web image classification (Huiskes & Lew, 2008) |
| Birdsong | 4,998 | 38 | 13 | 2.18 | 0% | tabular | bird song classification (Briggs et al., 2012) |
| Malagasy | 5,303 | 384 | 44 | 8.35 | 0.04% | tabular | POS Tagging (Garrette & Baldridge, 2013) |
| Soccer Player | 17,472 | 279 | 171 | 2.09 | 0% | tabular | automatic face naming (Zeng et al., 2013) |
| Italian | 21,878 | 519 | 90 | 1.60 | 0% | tabular | POS Tagging (Johan et al., 2009) |
| Yahoo! News | 22,991 | 163 | 219 | 1.91 | 0% | tabular | automatic face naming (Guillaumin et al., 2010) |
| English | 24,000 | 300 | 45 | 1.19 | 0.97% | tabular | POS Tagging (Zhou et al., 2018) |
| PLCIFAR10-Aggregate (Ours) | 50,000 | 3,072 | 10 | 4.87 | 0.13% | image | image classification (Krizhevsky & Hinton, 2009) |
| PLCIFAR10-Vaguest (Ours) | 50,000 | 3,072 | 10 | 3.49 | 17.56% | image | image classification (Krizhevsky & Hinton, 2009) |

### B.1  MORE DETAILS OF PLCIFAR10

For each example, we have a list of lists, where each sublist contains partial labels given by a single annotator. Each image was resized to $256 \times 256$ for easy annotation. We also imposed several requirements on the annotation task to ensure the quality of the annotations. First, we asked the annotators to choose labels for all ten images. Second, we did not allow the same annotator to select the same label for more than a threshold number of images. Third, we did not allow annotators to select too many partial labels more than a threshold for a single image since the annotation may contain little supervision information. We set the thresholds to 6 for a HIT of 10 images. Crowdworkers were involved in the data collection process. We paid $0.02 for a HIT, where $0.01 was given to the crowdworkers and $0.01 was given to the MTurk platform.

### B.2  MORE INFORMATION OF TABULAR DATASETS

For face age estimation, we considered ten crowdsourced labels of age numbers along with the true label as partial labels for a given human face. For automatic face naming, we considered the names in the corresponding captions or subtitles as partial labels for a face cropped from an image. For object classification, we considered object classes as partial labels for a segmentation part in an image. For web image classification, we considered tags on a web page as partial labels for a given image. For bird song classification, we considered bird species appearing in a ten-second bird song fragment as partial labels for the singing syllables of the fragment. For POS tagging, we considered all possible POS tags as partial labels for a given word with its contexts.

## C  USAGE OF PLENCH

PLCIFAR10 is located in `plench/data/plcifar10` in the code package. Tabular datasets can be downloaded from `https://palm.seu.edu.cn/zhangml/Resources.htm#data`. Researchers can easily add newly developed algorithms to PLENCH to verify their effectiveness. We mainly need to inherit the `Algorithm` class and implement the `update` and `predict` functions for the new algorithm in `algorithms.py`. For

Table 6: Summary of benchmark algorithms.

| | |
|---|---|
| **Vanilla Deep PLL Algorithms** | PRODEN (Lv et al., 2020), CAVL (Zhang et al., 2022), POP (Xu et al., 2023b), ABS-MAE (Lv et al., 2024a), ABS-GCE (Lv et al., 2024a), EXP (Feng et al., 2020a), MCL-GCE (Feng et al., 2020a), MCL-MSE (Feng et al., 2020a), CC (Feng et al., 2020b), LWS (Wen et al., 2021), IDGP (Qiao et al., 2023a) |
| **Vanilla Deep CLL Algorithms** | PC (Ishida et al., 2017), Forward (Yu et al., 2018), NN (Ishida et al., 2019), GA (Ishida et al., 2019), SCL-EXP (Chou et al., 2020), SCL-NL (Chou et al., 2020), L-W (Gao & Zhang, 2021), OP-W (Liu et al., 2023) |
| **Holistic Deep PLL Algorithms** | VALEN (Xu et al., 2021), PiCO (Wang et al., 2022b), ABLE (Xia et al., 2022), CRDPLL (Wu et al., 2022), DIRK (Wu et al., 2024) |
| **Deep Noisy PLL Algorithms** | FREDIS (Qiao et al., 2023b), ALIM (Xu et al., 2023a), PiCO+ (Wang et al., 2024a) |

example, to implement CAVL (Zhang et al., 2022), we can write the following code:

```python
class CAVL(Algorithm):
    def __init__(self, input_shape, train_givenY, hparams):
        super(CAVL, self).__init__(input_shape, train_givenY, hparams)
        self.featurizer = networks.Featurizer(input_shape, self.hparams)
        self.classifier = networks.Classifier(
            self.featurizer.n_outputs,
            self.num_classes)

        self.network = nn.Sequential(self.featurizer, self.classifier)
        self.optimizer = torch.optim.Adam(
            self.network.parameters(),
            lr=self.hparams["lr"],
            weight_decay=self.hparams['weight_decay']
        )
        train_givenY = torch.from_numpy(train_givenY)
        tempY = train_givenY.sum(dim=1).unsqueeze(1).repeat(1, train_givenY.shape[1])
        label_confidence = train_givenY.float()/tempY
        self.label_confidence = label_confidence
        self.label_confidence = self.label_confidence.double()

    def update(self, minibatches):
        _, x, strong_x, partial_y, _, index = minibatches
        loss = self.rc_loss(self.predict(x), index)
        self.optimizer.zero_grad()
        loss.backward()
        self.optimizer.step()
        self.confidence_update(x, partial_y, index)
        return {'loss': loss.item()}

    def rc_loss(self, outputs, index):
        device = "cuda" if index.is_cuda else "cpu"
        self.label_confidence = self.label_confidence.to(device)
        logsm_outputs = F.log_softmax(outputs, dim=1)
        final_outputs = logsm_outputs * self.label_confidence[index, :]
        average_loss = - ((final_outputs).sum(dim=1)).mean()
        return average_loss

    def predict(self, x):
        return self.network(x)

    def confidence_update(self, batchX, batchY, batch_index):
        with torch.no_grad():
            batch_outputs = self.predict(batchX)
            cav = (batch_outputs*torch.abs(1-batch_outputs))*batchY
            cav_pred = torch.max(cav,dim=1)[1]
            gt_label = F.one_hot(cav_pred,batchY.shape[1])
            self.label_confidence[batch_index,:] = gt_label.double()
```

After implementing the algorithmic code, we can specify the necessary hyperparameters for the algorithm in `hparams_registry.py`. Then we can train the model using the new algorithm with the following script:

```
python -m \textsc{Plench}.train --data_dir=your_data_path --algorithm CAVL \
--dataset PLCIFAR10_Aggregate --output_dir=your_output_path --steps 60000
```

# D  RELATED WORK

There are three main categories of deep PLL algorithms, including identification-based strategies, averaging-based strategies, and data-generation-based strategies. Identification-based strategies try to find the true label from the candidate label set and train a classifier simultaneously (Yao et al.,

2020; Lv et al., 2020; Wang & Zhang, 2020; Zhang et al., 2022; Wang & Zhang, 2022; Li et al., 2023; Xu et al., 2023b; Gong et al., 2022; He et al., 2024; Tian et al., 2024; Lv et al., 2024b). Averaging-based strategies consider equal contributions from all candidate labels and use the average modeling output as the final output (Lv et al., 2024a). Data-generation-based strategies model the generative relationship between partial labels and true labels, and derive loss functions with theoretical guarantees (Feng et al., 2020b;a; Wen et al., 2021; Qiao et al., 2023a). In addition, many works use strong representation learning techniques to improve model performance based on these basic strategies, such as contrastive learning and consistency regularization (Wang et al., 2022b; Wu et al., 2022; Xia et al., 2022; Wu et al., 2024). There are some work that also handles data annotations with multiple labels (Khetan et al., 2018; Peterson et al., 2019; Collins et al., 2022; Schmarje et al., 2022; Gao et al., 2022; Goswami et al., 2023) .

# E  DETAILS OF DEEP PLL ALGORITHMS

In this section, we first give detailed descriptions of the PLL algorithms used in this paper. Then, we provide the hyperparameter configurations for all the algorithms. Table 6 shows the summary of the deep PLL algorithms included in this paper.

## E.1  DESCRIPTIONS OF ALGORITHMS

The vanilla deep PLL algorithms include:

- PRODEN (Lv et al., 2020): An identification-based strategy that uses self-training to progressively estimate the true label distribution from the candidate label set.
- CAVL (Zhang et al., 2022): An identification-based strategy that uses the class activation value to directly identify the true label from the candidate label set.
- POP (Xu et al., 2023b): An identification-based strategy that progressively filters out false positive labels from the candidate label set based on PRODEN.
- ABS-MAE (Lv et al., 2024a): An averaging-based strategy using the mean absolute error (MAE) loss function.
- ABS-GCE (Lv et al., 2024a): An averaging-based strategy using the generalized cross-entropy (GCE) loss function.
- CC (Feng et al., 2020b): A data-generation-based strategy using a classifier-consistent loss function based on the uniform distribution assumption.
- EXP (Feng et al., 2020a): A data-generation-based strategy using exponential loss under the uniform distribution assumption.
- MCL-GCE (Feng et al., 2020a): A data-generation-based strategy using generalized cross-entropy (GCE) loss under the uniform distribution assumption.
- MCL-MSE (Feng et al., 2020a): A data-generation-based strategy using mean squared error (MSE) loss based on the uniform distribution assumption.
- LWS (Wen et al., 2021): A data-generation-based strategy that uses a leveraged weighted loss function to account for losses from both candidate and non-candidate labels.
- IDGP (Qiao et al., 2023a): A data-generation-based strategy that performs Maximum A Posterior (MAP) using a decomposed probability distribution model.

The vanilla deep CLL algorithms include

- PC (Ishida et al., 2017): A risk-consistent CLL algorithm using the pairwise-comparison loss based on the uniform distribution assumption.
- Forward (Yu et al., 2018): A classifier-consistent CLL algorithm using a transition matrix to model the complementary-label generation process based on the biased distribution assumption.
- NN (Ishida et al., 2019): A risk-consistent CLL algorithm using a non-negative risk estimator based on the uniform distribution assumption.
- GA (Ishida et al., 2019): A risk-consistent CLL algorithm using the gradient ascent technique based on the uniform distribution assumption.
- SCL-EXP (Chou et al., 2020): A discriminative CLL algorithm using exponential loss.
- SCL-NL (Chou et al., 2020): A discriminative CLL algorithm using negative loss.
- L-W (Gao & Zhang, 2021): A discriminative CLL algorithm using weighted loss.
- OP-W (Liu et al., 2023): A classifier-consistent CLL algorithm by using the opposite number of logits to compute the loss.

Table 7: Classification accuracy (mean±std) of each algorithm on Lost with different model selection criteria.

| Algorithm | Venue | w/ CR | w/ AA | w/ OA | w/ OA & ES |
|---|---|---|---|---|---|
| PRODEN | ICML 2020 (Lv et al., 2020) | 71.33±2.45 | 70.09±2.33 | 71.33±2.41 | 71.33±2.29 |
| CAVL | ICLR 2022 (Zhang et al., 2022) | 60.00±3.30 | 58.41±2.99 | 64.42±2.80 | 61.95±2.40 |
| POP | ICML 2023 (Xu et al., 2023b) | 69.38±1.96 | 69.38±1.96 | 70.44±2.45 | 73.10±2.16 |
| ABS-MAE | TPAMI 2024 (Lv et al., 2024a) | 68.50±2.70 | 67.43±2.10 | 68.85±3.07 | 69.03±2.99 |
| ABS-GCE | TPAMI 2024 (Lv et al., 2024a) | 62.65±2.64 | 62.30±2.04 | 62.48±2.66 | 67.43±2.68 |
| EXP | ICML 2020 (Feng et al., 2020a) | 71.33±1.27 | 70.27±2.32 | 72.21±1.59 | 70.97±1.26 |
| MCL-GCE | ICML 2020 (Feng et al., 2020a) | 60.71±2.41 | 58.94±2.62 | 61.95±1.99 | 65.49±2.14 |
| MCL-MSE | ICML 2020 (Feng et al., 2020a) | 58.58±2.26 | 59.47±2.71 | 58.76±2.52 | 59.47±3.06 |
| CC | NeurIPS 2020 (Feng et al., 2020b) | 71.33±2.60 | 70.62±2.92 | 72.57±2.80 | 72.21±2.42 |
| LWS | ICML 2021 (Wen et al., 2021) | 72.92±3.59 | 71.33±3.75 | 72.21±2.41 | 74.34±2.68 |
| IDGP | ICLR 2023 (Qiao et al., 2023a) | 69.73±2.87 | 70.97±1.59 | 75.04±1.61 | 71.68±2.39 |
| PC | NeurIPS 2017 (Ishida et al., 2017) | 66.37±2.54 | 63.01±2.89 | 66.37±2.33 | 67.96±2.29 |
| Forward | ECCV 2018 (Yu et al., 2018) | 71.68±3.70 | 71.50±3.77 | 73.27±2.91 | 72.74±3.51 |
| NN | ICML 2019 (Ishida et al., 2019) | 26.73±1.04 | 17.52±1.49 | 22.48±1.24 | 26.37±1.04 |
| GA | ICML 2019 (Ishida et al., 2019) | 18.94±3.40 | 13.63±2.57 | 6.73±1.14 | 20.00±2.06 |
| SCL-EXP | ICML 2020 (Chou et al., 2020) | 69.73±2.50 | 70.80±2.66 | 73.81±1.95 | 73.81±1.85 |
| SCL-NL | ICML 2020 (Chou et al., 2020) | 71.68±2.54 | 70.80±3.03 | 74.69±2.97 | 72.74±1.93 |
| L-W | ICML 2021 (Gao & Zhang, 2021) | 63.72±2.36 | 63.36±1.19 | 63.01±2.81 | 65.31±2.33 |
| OP-W | AISTATS 2023 (Liu et al., 2023) | 73.10±3.14 | 62.30±2.61 | 76.28±2.78 | 76.99±2.68 |
| VALEN | NeurIPS 2021 (Xu et al., 2021) | 66.02±2.43 | 65.31±2.55 | 64.42±2.98 | 66.19±3.32 |

The holistic deep PLL algorithms:

- VALEN (Xu et al., 2021): An identification-based strategy that uses variational inference to estimate the true label distribution.
- PiCO (Wang et al., 2022b): A PLL algorithm that uses the supervised contrastive learning module to improve model performance.
- ABLE (Xia et al., 2022): A PLL algorithm that uses an ambiguity-induced contrastive learning module to improve model performance.
- CRDPLL (Wu et al., 2022): A PLL algorithm that uses consistency regularization to improve model performance.
- DIRK (Wu et al., 2024): A PLL algorithm that uses knowledge distillation and contrastive learning to improve model performance.

The deep noisy PLL algorithms include:

- FREDIS (Qiao et al., 2023b): A noisy PLL algorithm that filters out false positive labels while including false negative labels.
- ALIM (Xu et al., 2023a): A noisy PLL algorithm that uses a weighted sum of the labeling confidence of candidate and non-candidate label sets as the target. We assume that the true noise rate is accessible.
- PiCO+ (Wang et al., 2024a): A noisy PLL algorithm using distance-based clean sample selection semi-supervised contrastive learning. We assume that the true noise rate is accessible.

### E.2 IMPLEMENTATION DETAILS

All the algorithms were implemented in PyTorch (Paszke et al., 2019) and all experiments were conducted with a single NVIDIA Tesla V100 GPU. We used the Adam optimizer (Kingma & Ba, 2015). We ran 60,000 iterations for the image datasets, 20,000 iterations for the Soccer Player, Italian, Yahoo! News, and English datasets, and 10,000 iterations for the other datasets. We recorded the performance on validation and test sets per 1,000 iterations. We used three random data splits for PLCIFAR10 and five random data splits for tabular datasets. For each data split, we selected 20 random hyperparameter configurations from a given pool. Table 8 shows the details of the hyperparameter configurations for all algorithms.

Table 8: Hyperparameters, their default values, and distributions for random search.

| Condition | Parameter | Default Value | Random Distribution |
|---|---|---|---|
| ResNet | learning rate | 0.001 | $10^{\text{Uniform}(-4.5,-2.5)}$ |
| | batch size | 256 | $2^{\text{Uniform}(6,9)}$ |
| | weight decay | 0.00001 | $10^{\text{Uniform}(-6,-3)}$ |
| MLP | learning rate | 0.001 | $10^{\text{Uniform}(-4.5,-2.5)}$ |
| | batch size | 128 | $2^{\text{Uniform}(5,8)}$ |
| | weight decay | 0.00001 | $10^{\text{Uniform}(-6,-3)}$ |
| POP | window size | 5 | $\text{RandomChoice}([3,4,5,6,7])$ |
| | warm up epoch | 20 | $\text{RandomChoice}([10,15,20])$ |
| | initial threshold | 0.001 | $10^{\text{Uniform}(-4.5,-2.5)}$ |
| | step size | 0.001 | $10^{\text{Uniform}(-4.5,-2.5)}$ |
| ABS-GCE | q | 0.7 | 0.7 |
| MCL-GCE | q | 0.7 | 0.7 |
| LWS | leveraged weight | 2 | $\text{RandomChoice}([1,2])$ |
| IDGP | warm up epoch | 10 | $\text{RandomChoice}([5,10,15,20])$ |
| VALEN | warm up iteration | 5000 | $\text{RandomChoice}([1000,2000,3000,4000,5000])$ |
| | number of nearest neighbors | 3 | $\text{RandomChoice}([3,4])$ |
| PiCO | warm up iteration | 178 | 178 |
| | size of output representation | 128 | $\text{RandomChoice}([64,128,256])$ |
| | size of the queue | 8192 | $\text{RandomChoice}([4096,8192])$ |
| | model update momentum | 0.999 | $\text{RandomChoice}([0.9,0.999])$ |
| | prototype calculation momentum | 0.99 | $\text{RandomChoice}([0.99,0.9])$ |
| | weight of the contrastive loss | 0.5 | $\text{RandomChoice}([0.5,1.0])$ |
| ABLE | size of output representation | 128 | $\text{RandomChoice}([64,128,256])$ |
| | weight of the ambiguity-induced loss | 1 | $\text{RandomChoice}([0.5,1,2])$ |
| | temperature | 0.07 | $\text{RandomChoice}([0.03,0.05,0.07,0.09])$ |
| CRDPLL | weight of the consistency loss | 1 | 1 |
| DIRK | teacher model update momentum | 0.99 | $\text{RandomChoice}([0.5,0.9,0.99])$ |
| FREDIS | initial refinement threshold | 0.000001 | 0.000001 |
| | initial disambiguation threshold | 1 | 1 |
| | step size of refinement threshold | 0.000001 | $0.1^{\text{RandomChoice}[6,5,4,3,2,1]}$ |
| | step size of disambiguation threshold | 0.000001 | $0.1^{\text{RandomChoice}[6,5,4,3,2,1]}$ |
| | number of changed entries | 500 | $\text{RandomChoice}([50,100,300,500,800,1000])$ |
| | time of the number of changed entries | 2 | $\text{RandomChoice}([1,2,3,4,5])$ |
| | epoch of updating intervals | 20 | $\text{RandomChoice}([10,20])$ |
| | weight of the consistency loss | 10 | $\text{RandomChoice}([1,10])$ |
| | weight of the supervised loss | 1 | $\text{RandomChoice}([0.001,0.01,0.1,1])$ |
| ALIM | warm up iteration | 178 | 178 |
| | size of output representation | 128 | $\text{RandomChoice}([64,128,256])$ |
| | size of the queue | 8192 | $\text{RandomChoice}([4096,8192])$ |
| | model update momentum | 0.999 | $\text{RandomChoice}([0.9,0.999])$ |
| | prototype calculation momentum | 0.99 | $\text{RandomChoice}([0.99,0.9])$ |
| | weight of the contrastive loss | 0.5 | $\text{RandomChoice}([0.5,1.0])$ |
| | starting epoch of denoising | 40 | $\text{RandomChoice}([20,40,80,100,140])$ |
| | weight of the mixup loss | 1 | 1 |
| PiCO+ | warm up iteration | 250 | $\text{RandomChoice}([1,250])$ |
| | size of output representation | 128 | $\text{RandomChoice}([64,128,256])$ |
| | size of the queue | 8192 | $\text{RandomChoice}([4096,8192])$ |
| | model update momentum | 0.999 | $\text{RandomChoice}([0.9,0.999])$ |
| | prototype calculation momentum | 0.99 | $\text{RandomChoice}([0.99,0.9])$ |
| | weight of the contrastive loss | 0.5 | $\text{RandomChoice}([0.5,1.0])$ |
| | selection ratio for clean sample | 1 | $\text{RandomChoice}([0.6,0.8,0.95,0.99,1])$ |
| | start iteration of $k$NN augmentation | 5000 | $\text{RandomChoice}([2,100,5000])$ |
| | number of nearest neighbors | 16 | $\text{RandomChoice}([8,16])$ |
| | temperature of label guessing | 0.07 | $\text{RandomChoice}([0.1,0.07])$ |
| | weight for the losses of unreliable examples | 0.1 | $\text{RandomChoice}([0.1,0.5])$ |
| | weight for the losses of mixup loss | 2 | $\text{RandomChoice}([2,3,5])$ |

## F    DETAILS OF EXPERIMENTAL RESULTS

Table 7 and Tables 9 to 16 show the experimental results on tabular datasets.

Table 9: Classification accuracy (mean±std) of each algorithm on MSRCv2 with different model selection criteria.

| Algorithm | Venue | w/ CR | w/ AA | w/ OA | w/ OA & ES |
|---|---|---|---|---|---|
| PRODEN | ICML 2020 (Lv et al., 2020) | 52.50±0.83 | 53.41±1.11 | 51.82±0.65 | 52.39±1.16 |
| CAVL | ICLR 2022 (Zhang et al., 2022) | 46.14±0.84 | 45.00±1.26 | 48.64±1.56 | 46.14±1.63 |
| POP | ICML 2023 (Xu et al., 2023b) | 51.36±0.92 | 52.05±0.73 | 52.61±1.10 | 54.09±1.01 |
| ABS-MAE | TPAMI 2024 (Lv et al., 2024a) | 45.68±1.82 | 44.09±1.80 | 46.48±1.91 | 48.18±1.47 |
| ABS-GCE | TPAMI 2024 (Lv et al., 2024a) | 48.30±1.80 | 46.93±1.37 | 49.55±1.31 | 50.34±1.11 |
| EXP | ICML 2020 (Feng et al., 2020a) | 44.77±0.73 | 45.57±0.94 | 46.14±1.36 | 45.57±1.26 |
| MCL-GCE | ICML 2020 (Feng et al., 2020a) | 44.43±1.16 | 44.77±1.19 | 47.16±0.28 | 46.93±1.10 |
| MCL-MSE | ICML 2020 (Feng et al., 2020a) | 45.68±1.33 | 33.64±2.08 | 43.75±1.27 | 45.80±1.79 |
| CC | NeurIPS 2020 (Feng et al., 2020b) | 52.16±1.32 | 52.27±1.51 | 52.27±1.60 | 53.07±1.25 |
| LWS | ICML 2021 (Wen et al., 2021) | 47.50±0.80 | 46.59±0.53 | 49.77±1.35 | 49.32±1.03 |
| IDGP | ICLR 2023 (Qiao et al., 2023a) | 52.61±0.81 | 50.45±2.07 | 52.05±1.14 | 52.84±1.56 |
| PC | NeurIPS 2017 (Ishida et al., 2017) | 41.82±1.33 | 43.98±2.02 | 44.09±0.73 | 44.89±1.24 |
| Forward | ECCV 2018 (Yu et al., 2018) | 49.89±1.71 | 51.59±1.64 | 51.14±0.53 | 50.57±1.80 |
| NN | ICML 2019 (Ishida et al., 2019) | 18.86±1.79 | 16.82±1.33 | 16.70±0.83 | 20.23±1.14 |
| GA | ICML 2019 (Ishida et al., 2019) | 12.95±1.01 | 13.52±0.67 | 10.91±0.74 | 14.55±1.22 |
| SCL-EXP | ICML 2020 (Chou et al., 2020) | 48.86±1.41 | 49.89±1.20 | 48.98±1.70 | 49.32±1.67 |
| SCL-NL | ICML 2020 (Chou et al., 2020) | 51.25±1.02 | 51.70±1.41 | 49.32±1.11 | 51.36±1.24 |
| L-W | ICML 2021 (Gao & Zhang, 2021) | 44.77±1.63 | 41.93±0.74 | 45.34±0.90 | 44.55±0.89 |
| OP-W | AISTATS 2023 (Liu et al., 2023) | 49.89±0.98 | 27.50±9.01 | 51.59±1.55 | 51.93±1.42 |
| VALEN | NeurIPS 2021 (Xu et al., 2021) | 48.30±1.43 | 49.09±1.29 | 48.41±0.93 | 49.89±0.81 |

Table 10: Classification accuracy (mean±std) of each algorithm on Mirflickr with different model selection criteria.

| Algorithm | Venue | w/ CR | w/ AA | w/ OA | w/ OA & ES |
|---|---|---|---|---|---|
| PRODEN | ICML 2020 (Lv et al., 2020) | 66.19±1.61 | 65.40±1.00 | 66.47±1.63 | 66.69±1.29 |
| CAVL | ICLR 2022 (Zhang et al., 2022) | 63.88±0.69 | 64.75±1.40 | 62.73±1.69 | 65.83±1.17 |
| POP | ICML 2023 (Xu et al., 2023b) | 65.40±1.18 | 64.53±0.78 | 66.62±1.05 | 66.40±1.28 |
| ABS-MAE | TPAMI 2024 (Lv et al., 2024a) | 63.60±0.98 | 58.49±1.92 | 63.81±1.01 | 66.04±1.07 |
| ABS-GCE | TPAMI 2024 (Lv et al., 2024a) | 53.96±1.71 | 53.02±0.78 | 54.17±0.90 | 58.27±1.24 |
| EXP | ICML 2020 (Feng et al., 2020a) | 64.46±1.39 | 57.55±3.13 | 64.82±1.32 | 63.53±1.76 |
| MCL-GCE | ICML 2020 (Feng et al., 2020a) | 55.40±1.35 | 53.17±1.29 | 54.10±0.89 | 56.62±0.22 |
| MCL-MSE | ICML 2020 (Feng et al., 2020a) | 52.23±1.01 | 35.04±5.62 | 55.68±1.25 | 57.27±1.59 |
| CC | NeurIPS 2020 (Feng et al., 2020b) | 65.04±1.16 | 63.67±1.37 | 64.32±0.65 | 66.98±1.03 |
| LWS | ICML 2021 (Wen et al., 2021) | 64.96±1.30 | 38.78±5.29 | 65.04±1.23 | 65.54±1.77 |
| IDGP | ICLR 2023 (Qiao et al., 2023a) | 66.12±2.07 | 68.35±1.22 | 68.13±0.98 | 69.06±0.91 |
| PC | NeurIPS 2017 (Ishida et al., 2017) | 62.45±1.14 | 58.78±1.41 | 62.37±1.07 | 63.02±1.61 |
| Forward | ECCV 2018 (Yu et al., 2018) | 65.32±1.45 | 64.68±2.16 | 64.96±1.19 | 65.54±1.33 |
| NN | ICML 2019 (Ishida et al., 2019) | 19.93±1.78 | 9.28±1.19 | 18.06±1.22 | 24.96±1.52 |
| GA | ICML 2019 (Ishida et al., 2019) | 21.37±0.52 | 11.37±4.01 | 17.84±0.39 | 21.51±1.86 |
| SCL-EXP | ICML 2020 (Chou et al., 2020) | 63.31±1.29 | 59.42±4.38 | 64.60±1.03 | 65.32±0.86 |
| SCL-NL | ICML 2020 (Chou et al., 2020) | 63.60±1.13 | 63.74±1.17 | 67.12±1.15 | 66.83±0.62 |
| L-W | ICML 2021 (Gao & Zhang, 2021) | 57.55±1.31 | 52.09±3.57 | 57.34±2.71 | 60.07±2.18 |
| OP-W | AISTATS 2023 (Liu et al., 2023) | 64.32±1.47 | 15.32±9.24 | 64.53±1.24 | 65.83±1.21 |
| VALEN | NeurIPS 2021 (Xu et al., 2021) | 58.56±1.69 | 59.42±1.10 | 58.78±1.70 | 60.58±0.94 |

Table 11: Classification accuracy (mean±std) of each algorithm on Birdsong with different model selection criteria.

| Algorithm | Venue | w/ CR | w/ AA | w/ OA | w/ OA & ES |
|---|---|---|---|---|---|
| PRODEN | ICML 2020 (Lv et al., 2020) | 71.64±0.61 | 70.16±0.88 | 71.44±0.65 | 72.32±0.89 |
| CAVL | ICLR 2022 (Zhang et al., 2022) | 65.96±0.74 | 66.04±0.36 | 66.56±0.41 | 67.08±0.68 |
| POP | ICML 2023 (Xu et al., 2023b) | 71.80±0.98 | 69.96±0.58 | 72.60±0.92 | 72.24±0.73 |
| ABS-MAE | TPAMI 2024 (Lv et al., 2024a) | 66.20±0.83 | 63.48±1.17 | 68.20±0.70 | 68.28±0.72 |
| ABS-GCE | TPAMI 2024 (Lv et al., 2024a) | 69.04±0.55 | 67.96±0.79 | 69.44±0.56 | 70.16±0.94 |
| EXP | ICML 2020 (Feng et al., 2020a) | 64.16±0.74 | 62.64±1.22 | 65.20±0.70 | 65.24±1.09 |
| MCL-GCE | ICML 2020 (Feng et al., 2020a) | 65.92±1.40 | 64.96±0.95 | 64.92±0.83 | 67.56±0.80 |
| MCL-MSE | ICML 2020 (Feng et al., 2020a) | 65.64±0.92 | 55.48±1.87 | 67.04±0.52 | 66.68±0.32 |
| CC | NeurIPS 2020 (Feng et al., 2020b) | 70.88±0.85 | 68.68±0.65 | 72.24±0.57 | 71.96±0.55 |
| LWS | ICML 2021 (Wen et al., 2021) | 66.04±0.43 | 56.48±5.22 | 65.84±0.61 | 66.04±0.58 |
| IDGP | ICLR 2023 (Qiao et al., 2023a) | 72.48±0.61 | 69.72±0.50 | 73.40±0.64 | 73.24±0.62 |
| PC | NeurIPS 2017 (Ishida et al., 2017) | 68.92±0.33 | 66.56±1.13 | 70.08±0.52 | 69.88±0.72 |
| Forward | ECCV 2018 (Yu et al., 2018) | 69.84±0.77 | 69.80±0.41 | 70.00±0.50 | 69.88±0.61 |
| NN | ICML 2019 (Ishida et al., 2019) | 18.68±1.28 | 16.04±0.80 | 18.00±0.70 | 20.72±0.91 |
| GA | ICML 2019 (Ishida et al., 2019) | 20.08±1.24 | 20.04±1.14 | 13.24±0.73 | 19.84±1.15 |
| SCL-EXP | ICML 2020 (Chou et al., 2020) | 70.12±0.61 | 68.84±0.93 | 70.72±0.38 | 71.08±0.61 |
| SCL-NL | ICML 2020 (Chou et al., 2020) | 70.32±0.62 | 70.16±0.95 | 70.28±0.50 | 70.28±0.78 |
| L-W | ICML 2021 (Gao & Zhang, 2021) | 62.48±0.71 | 61.80±0.83 | 66.48±0.82 | 67.60±0.81 |
| OP-W | AISTATS 2023 (Liu et al., 2023) | 69.60±0.66 | 51.72±9.37 | 69.60±0.74 | 71.80±0.91 |
| VALEN | NeurIPS 2021 (Xu et al., 2021) | 66.76±1.62 | 66.76±1.07 | 66.76±0.67 | 68.44±1.00 |

Table 12: Classification accuracy (mean±std) of each algorithm on Malagasy with different model selection criteria.

| Algorithm | Venue | w/ CR | w/ AA | w/ OA | w/ OA & ES |
|---|---|---|---|---|---|
| PRODEN | ICML 2020 (Lv et al., 2020) | 67.19±0.69 | 66.93±2.41 | 71.83±0.81 | 71.15±0.58 |
| CAVL | ICLR 2022 (Zhang et al., 2022) | 65.46±1.97 | 64.71±1.21 | 68.55±0.98 | 69.34±1.08 |
| POP | ICML 2023 (Xu et al., 2023b) | 62.52±1.89 | 65.39±2.12 | 70.47±0.65 | 70.43±0.81 |
| ABS-MAE | TPAMI 2024 (Lv et al., 2024a) | 58.46±3.15 | 54.39±3.29 | 66.52±0.62 | 66.06±0.44 |
| ABS-GCE | TPAMI 2024 (Lv et al., 2024a) | 60.64±1.38 | 59.70±2.10 | 63.47±1.60 | 65.76±1.13 |
| EXP | ICML 2020 (Feng et al., 2020a) | 0.23±0.10 | 0.23±0.10 | 0.23±0.10 | 0.23±0.10 |
| MCL-GCE | ICML 2020 (Feng et al., 2020a) | 0.23±0.10 | 0.23±0.10 | 0.23±0.10 | 0.23±0.10 |
| MCL-MSE | ICML 2020 (Feng et al., 2020a) | 0.23±0.10 | 0.23±0.10 | 0.23±0.10 | 0.23±0.10 |
| CC | NeurIPS 2020 (Feng et al., 2020b) | 57.82±0.78 | 60.53±2.32 | 70.66±0.66 | 71.41±1.00 |
| LWS | ICML 2021 (Wen et al., 2021) | 57.74±1.33 | 45.50±4.37 | 63.54±1.56 | 65.08±1.02 |
| IDGP | ICLR 2023 (Qiao et al., 2023a) | 69.94±0.70 | 67.50±1.45 | 70.06±0.62 | 70.70±0.83 |
| PC | NeurIPS 2017 (Ishida et al., 2017) | 67.01±1.36 | 62.52±1.38 | 69.15±1.08 | 69.38±0.82 |
| Forward | ECCV 2018 (Yu et al., 2018) | 61.51±2.55 | 62.79±2.60 | 69.72±0.44 | 69.38±0.79 |
| NN | ICML 2019 (Ishida et al., 2019) | 19.89±1.75 | 14.43±2.04 | 20.34±1.00 | 25.54±0.75 |
| GA | ICML 2019 (Ishida et al., 2019) | 18.83±1.51 | 19.47±1.68 | 9.04±0.65 | 19.55±1.62 |
| SCL-EXP | ICML 2020 (Chou et al., 2020) | 61.73±2.40 | 57.48±1.38 | 69.76±0.36 | 69.72±0.87 |
| SCL-NL | ICML 2020 (Chou et al., 2020) | 57.36±0.69 | 61.77±2.34 | 68.66±1.05 | 68.89±0.62 |
| L-W | ICML 2021 (Gao & Zhang, 2021) | 57.33±1.74 | 58.38±2.46 | 64.71±1.11 | 66.70±1.03 |
| OP-W | AISTATS 2023 (Liu et al., 2023) | 58.95±0.77 | 0.64±0.18 | 70.21±0.39 | 69.34±0.74 |
| VALEN | NeurIPS 2021 (Xu et al., 2021) | 65.76±1.23 | 60.23±2.71 | 68.78±0.75 | 68.63±0.64 |

Table 13: Classification accuracy (mean±std) of each algorithm on Soccer Player with different model selection criteria.

| Algorithm | Venue | w/ CR | w/ AA | w/ OA | w/ OA & ES |
|---|---|---|---|---|---|
| PRODEN | ICML 2020 (Lv et al., 2020) | 54.98±0.41 | 54.65±0.55 | 55.02±0.52 | 55.38±0.49 |
| CAVL | ICLR 2022 (Zhang et al., 2022) | 51.89±1.08 | 51.52±1.09 | 51.26±1.04 | 51.67±1.02 |
| POP | ICML 2023 (Xu et al., 2023b) | 54.74±0.55 | 54.57±0.62 | 54.82±0.56 | 55.02±0.40 |
| ABS-MAE | TPAMI 2024 (Lv et al., 2024a) | 48.90±0.62 | 48.75±0.54 | 48.90±0.62 | 48.90±0.62 |
| ABS-GCE | TPAMI 2024 (Lv et al., 2024a) | 55.33±0.48 | 48.95±0.55 | 55.70±0.68 | 55.90±0.67 |
| EXP | ICML 2020 (Feng et al., 2020a) | 49.05±0.55 | 48.78±0.54 | 49.04±0.52 | 48.92±0.50 |
| MCL-GCE | ICML 2020 (Feng et al., 2020a) | 53.64±0.57 | 49.51±0.53 | 53.56±0.62 | 53.40±0.57 |
| MCL-MSE | ICML 2020 (Feng et al., 2020a) | 52.22±0.55 | 51.41±0.28 | 53.03±0.59 | 53.65±0.53 |
| CC | NeurIPS 2020 (Feng et al., 2020b) | 54.15±0.64 | 54.86±0.33 | 54.93±0.38 | 54.71±0.42 |
| LWS | ICML 2021 (Wen et al., 2021) | 52.55±0.48 | 48.99±0.51 | 52.60±0.45 | 52.69±0.50 |
| IDGP | ICLR 2023 (Qiao et al., 2023a) | 54.39±0.71 | 54.16±0.58 | 54.97±0.42 | 55.13±0.68 |
| PC | NeurIPS 2017 (Ishida et al., 2017) | 54.85±0.76 | 48.75±0.54 | 54.70±0.72 | 54.45±0.64 |
| Forward | ECCV 2018 (Yu et al., 2018) | 50.35±0.93 | 49.93±1.01 | 50.57±0.86 | 50.47±0.95 |
| NN | ICML 2019 (Ishida et al., 2019) | 11.10±0.38 | 3.04±0.29 | 9.97±0.35 | 11.10±0.38 |
| GA | ICML 2019 (Ishida et al., 2019) | 6.06±0.46 | 4.95±1.17 | 5.06±0.24 | 6.02±0.50 |
| SCL-EXP | ICML 2020 (Chou et al., 2020) | 49.32±0.87 | 49.35±0.87 | 49.51±0.89 | 49.42±0.84 |
| SCL-NL | ICML 2020 (Chou et al., 2020) | 50.57±0.83 | 49.98±1.06 | 50.58±0.91 | 50.55±0.95 |
| L-W | ICML 2021 (Gao & Zhang, 2021) | 49.41±0.69 | 48.84±0.59 | 49.60±0.60 | 49.83±0.66 |
| OP-W | AISTATS 2023 (Liu et al., 2023) | 50.77±0.48 | 28.54±10.31 | 51.12±0.61 | 50.93±0.51 |
| VALEN | NeurIPS 2021 (Xu et al., 2021) | 52.13±0.54 | 51.97±0.52 | 52.15±0.54 | 52.30±0.59 |

Table 14: Classification accuracy (mean±std) of each algorithm on Italian with different model selection criteria.

| Algorithm | Venue | w/ CR | w/ AA | w/ OA | w/ OA & ES |
|---|---|---|---|---|---|
| PRODEN | ICML 2020 (Lv et al., 2020) | 68.89±0.44 | 69.32±0.60 | 71.80±0.95 | 72.98±0.66 |
| CAVL | ICLR 2022 (Zhang et al., 2022) | 68.83±0.96 | 68.12±0.72 | 70.71±0.45 | 70.62±0.60 |
| POP | ICML 2023 (Xu et al., 2023b) | 68.48±0.67 | 70.19±0.36 | 71.56±1.03 | 71.78±0.91 |
| ABS-MAE | TPAMI 2024 (Lv et al., 2024a) | 67.60±0.32 | 66.92±0.84 | 70.11±1.31 | 70.07±1.26 |
| ABS-GCE | TPAMI 2024 (Lv et al., 2024a) | 66.76±0.64 | 65.36±0.68 | 69.60±0.57 | 70.90±0.54 |
| EXP | ICML 2020 (Feng et al., 2020a) | 67.58±0.64 | 67.28±0.61 | 71.43±0.74 | 71.64±0.46 |
| MCL-GCE | ICML 2020 (Feng et al., 2020a) | 66.22±0.72 | 66.03±0.75 | 69.64±0.41 | 70.14±0.31 |
| MCL-MSE | ICML 2020 (Feng et al., 2020a) | 66.48±0.16 | 65.86±1.12 | 68.29±0.52 | 69.16±0.47 |
| CC | NeurIPS 2020 (Feng et al., 2020b) | 65.96±1.03 | 66.89±1.11 | 71.40±0.32 | 72.09±0.54 |
| LWS | ICML 2021 (Wen et al., 2021) | 71.65±0.45 | 58.70±5.74 | 73.36±0.50 | 73.68±0.73 |
| IDGP | ICLR 2023 (Qiao et al., 2023a) | 68.43±0.50 | 67.43±0.82 | 70.08±0.19 | 70.73±0.42 |
| PC | NeurIPS 2017 (Ishida et al., 2017) | 70.10±0.69 | 67.41±1.23 | 71.22±0.33 | 71.73±0.42 |
| Forward | ECCV 2018 (Yu et al., 2018) | 68.28±1.29 | 68.28±1.29 | 71.20±0.86 | 71.63±0.66 |
| NN | ICML 2019 (Ishida et al., 2019) | 32.92±1.92 | 17.62±5.35 | 23.78±0.57 | 33.75±1.28 |
| GA | ICML 2019 (Ishida et al., 2019) | 11.62±1.61 | 11.98±1.43 | 5.88±0.25 | 11.98±1.43 |
| SCL-EXP | ICML 2020 (Chou et al., 2020) | 67.34±0.31 | 67.34±0.31 | 69.03±0.54 | 69.08±0.69 |
| SCL-NL | ICML 2020 (Chou et al., 2020) | 67.59±0.33 | 67.59±0.33 | 68.82±0.69 | 70.33±1.22 |
| L-W | ICML 2021 (Gao & Zhang, 2021) | 67.43±0.45 | 68.78±0.43 | 68.90±0.20 | 69.56±0.33 |
| OP-W | AISTATS 2023 (Liu et al., 2023) | 65.70±1.10 | 1.11±0.38 | 71.04±1.06 | 71.30±1.04 |
| VALEN | NeurIPS 2021 (Xu et al., 2021) | 67.97±0.86 | 67.29±0.42 | 69.31±0.64 | 71.12±0.80 |

Table 15: Classification accuracy (mean±std) of each algorithm on Yahoo! News with different model selection criteria.

| Algorithm | Venue | w/ CR | w/ AA | w/ OA | w/ OA & ES |
|---|---|---|---|---|---|
| PRODEN | ICML 2020 (Lv et al., 2020) | 66.83±0.52 | 66.29±0.54 | 66.71±0.58 | 66.81±0.51 |
| CAVL | ICLR 2022 (Zhang et al., 2022) | 57.77±1.53 | 57.14±1.64 | 57.30±1.55 | 58.30±1.57 |
| POP | ICML 2023 (Xu et al., 2023b) | 66.42±0.74 | 65.76±0.46 | 66.02±0.57 | 66.17±0.64 |
| ABS-MAE | TPAMI 2024 (Lv et al., 2024a) | 55.70±1.53 | 48.53±3.62 | 56.11±1.63 | 56.15±1.61 |
| ABS-GCE | TPAMI 2024 (Lv et al., 2024a) | 58.03±0.43 | 57.79±0.43 | 58.54±0.95 | 58.48±0.36 |
| EXP | ICML 2020 (Feng et al., 2020a) | 50.37±2.06 | 47.01±3.14 | 50.13±2.11 | 50.44±2.05 |
| MCL-GCE | ICML 2020 (Feng et al., 2020a) | 53.91±0.34 | 53.91±0.34 | 53.99±0.50 | 54.13±0.34 |
| MCL-MSE | ICML 2020 (Feng et al., 2020a) | 54.49±0.64 | 50.46±1.07 | 52.12±0.70 | 54.74±0.70 |
| CC | NeurIPS 2020 (Feng et al., 2020b) | 67.02±0.32 | 65.43±0.56 | 66.49±0.45 | 66.69±0.40 |
| LWS | ICML 2021 (Wen et al., 2021) | 67.64±0.73 | 39.87±9.64 | 67.18±0.55 | 67.46±0.32 |
| IDGP | ICLR 2023 (Qiao et al., 2023a) | 66.10±0.63 | 65.66±0.69 | 62.55±0.75 | 65.99±0.31 |
| PC | NeurIPS 2017 (Ishida et al., 2017) | 58.19±0.41 | 54.07±0.79 | 58.22±0.46 | 58.17±0.39 |
| Forward | ECCV 2018 (Yu et al., 2018) | 50.70±1.25 | 50.56±1.33 | 50.62±1.40 | 50.70±1.25 |
| NN | ICML 2019 (Ishida et al., 2019) | 23.80±0.24 | 1.77±0.16 | 23.23±0.27 | 24.19±0.20 |
| GA | ICML 2019 (Ishida et al., 2019) | 13.30±0.31 | 13.30±0.31 | 12.91±0.57 | 13.30±0.31 |
| SCL-EXP | ICML 2020 (Chou et al., 2020) | 50.30±1.26 | 50.35±1.23 | 50.57±1.28 | 50.70±1.21 |
| SCL-NL | ICML 2020 (Chou et al., 2020) | 50.59±1.22 | 50.50±1.19 | 50.76±1.01 | 50.85±1.16 |
| L-W | ICML 2021 (Gao & Zhang, 2021) | 43.95±0.67 | 42.98±0.66 | 42.71±0.62 | 44.60±0.70 |
| OP-W | AISTATS 2023 (Liu et al., 2023) | 57.37±0.96 | 45.21±9.93 | 57.35±1.05 | 57.49±0.92 |
| VALEN | NeurIPS 2021 (Xu et al., 2021) | 56.97±0.44 | 56.98±0.45 | 56.02±0.62 | 57.82±0.50 |

Table 16: Classification accuracy (mean±std) of each algorithm on English with different model selection criteria.

| Algorithm | Venue | w/ CR | w/ AA | w/ OA | w/ OA & ES |
|---|---|---|---|---|---|
| PRODEN | ICML 2020 (Lv et al., 2020) | 74.01±0.32 | 73.78±0.13 | 73.62±0.15 | 73.70±0.26 |
| CAVL | ICLR 2022 (Zhang et al., 2022) | 73.71±0.35 | 73.58±0.35 | 74.15±0.50 | 74.19±0.25 |
| POP | ICML 2023 (Xu et al., 2023b) | 74.04±0.32 | 73.99±0.17 | 73.78±0.31 | 74.44±0.36 |
| ABS-MAE | TPAMI 2024 (Lv et al., 2024a) | 73.07±0.22 | 72.37±0.44 | 73.20±0.13 | 73.31±0.28 |
| ABS-GCE | TPAMI 2024 (Lv et al., 2024a) | 73.52±0.30 | 73.12±0.26 | 74.47±0.16 | 74.57±0.23 |
| EXP | ICML 2020 (Feng et al., 2020a) | 73.18±0.23 | 72.73±0.24 | 73.75±0.27 | 73.61±0.33 |
| MCL-GCE | ICML 2020 (Feng et al., 2020a) | 73.52±0.24 | 73.52±0.51 | 74.09±0.26 | 74.35±0.48 |
| MCL-MSE | ICML 2020 (Feng et al., 2020a) | 73.70±0.24 | 73.37±0.48 | 74.22±0.39 | 73.96±0.24 |
| CC | NeurIPS 2020 (Feng et al., 2020b) | 73.83±0.25 | 73.77±0.29 | 73.62±0.08 | 73.87±0.19 |
| LWS | ICML 2021 (Wen et al., 2021) | 73.31±0.34 | 64.11±3.99 | 73.98±0.38 | 73.91±0.48 |
| IDGP | ICLR 2023 (Qiao et al., 2023a) | 74.01±0.26 | 73.82±0.24 | 73.90±0.35 | 74.44±0.23 |
| PC | NeurIPS 2017 (Ishida et al., 2017) | 73.47±0.29 | 72.82±0.30 | 73.38±0.45 | 73.74±0.35 |
| Forward | ECCV 2018 (Yu et al., 2018) | 73.40±0.21 | 73.60±0.37 | 73.88±0.24 | 73.40±0.48 |
| NN | ICML 2019 (Ishida et al., 2019) | 54.18±0.89 | 53.82±0.76 | 39.25±1.43 | 54.18±0.89 |
| GA | ICML 2019 (Ishida et al., 2019) | 34.78±1.90 | 34.77±1.91 | 23.19±0.84 | 34.77±1.91 |
| SCL-EXP | ICML 2020 (Chou et al., 2020) | 73.84±0.24 | 73.44±0.40 | 74.08±0.26 | 73.84±0.25 |
| SCL-NL | ICML 2020 (Chou et al., 2020) | 74.01±0.22 | 73.78±0.44 | 74.04±0.31 | 74.17±0.24 |
| L-W | ICML 2021 (Gao & Zhang, 2021) | 73.84±0.44 | 73.13±0.54 | 74.02±0.39 | 74.14±0.29 |
| OP-W | AISTATS 2023 (Liu et al., 2023) | 73.88±0.39 | 31.83±15.01 | 73.97±0.35 | 74.05±0.36 |
| VALEN | NeurIPS 2021 (Xu et al., 2021) | 73.20±0.41 | 72.86±0.40 | 73.39±0.23 | 73.77±0.24 |

