# OpenReview forum: "Realistic Evaluation of Deep Partial-Label Learning Algorithms"
_ICLR.cc/2025/Conference — ICLR 2025 Spotlight_

### Official Review · Reviewer_GD1c · 2024-10-26

**Soundness:** 3
**Presentation:** 3
**Contribution:** 3
**Rating:** 8
**Confidence:** 4

**Summary:**

This paper presents three key contributions: (1) an investigation of the model selection problem in the partial-label learning (PLL) setting, proposing several metrics such as covering rate (CR), approximated accuracy (AA), oracle accuracy (OA), and OA with early stopping (ES); (2) the creation of a partial label dataset based on CIFAR10, named PLCIFAR10; and (3) a benchmarking of multiple algorithms across several datasets. Overall, this paper serves as a comprehensive benchmarking study and does so effectively.

**Strengths:**

Comprehensive and easy to follow.

**Weaknesses:**

See my comments in 'Question'.

**Questions:**

I have a few questions and suggestions that I believe could help improve the paper:

1) Could the authors comment on the differences and similarities between their work and [1,2]? Both papers appear to address benchmarking in the PLL setting and should be discussed to provide a clearer context.

2) My major concern is that one of the main contributions is the proposed model selection criterion, while there is no experimental results to support its validity. Could the authors provide experimental evidence showing that the proposed criteria (CR, AA, OA, OA w/ ES) lead to better performance on the test dataset? Specifically, for a given ML method, does the model selected with CR (or AA, OA, OA w/ ES) outperform the model without selection?

3) While I understand there is no universally best model selection criterion, would it be possible for the authors to include a numerical comparison showing how often each criterion (CR, AA, OA, OA w/ ES) achieves the best performance? For instance, in Table I, there are 27 algorithms. How many times does the best performance come from models selected with CR, AA, OA, or OA w/ ES?

4) To improve clarity, could the authors provide further explanation of $p(x,y)$ and $p(S|x,y)$ in Eq (4)?

5) I want to bring two particular papers [3,4] into the author's sight.

[1] Lars Schmarje, et al., "Is one annotation enough? A data-centric image classification benchmark for noisy and ambiguous label estimation," NeurIPS 2022.

[2] Mononito Goswami, et al., "AQuA: A Benchmarking Tool for Label Quality Assessment," NeurIPS 2023.

[3] Zhengqi Gao et al., "Learning from multiple annotator noisy labels via sample-wise label fusion," ECCV 2022.

[4] Ashish Khetan, et al., 'Learning from noisy singly-labeled data,' ICLR 2018.

---

> ### Author Response · Authors · 2024-11-22
> **Response To Reviewer GD1c**
>
> First of all, we are very grateful for your help and time in reviewing our submission. Below are the responses to your questions and comments.
>
> **Q1: Could the authors comment on the differences and similarities between their work and [1,2]? Both papers appear to address benchmarking in the PLL setting and should be discussed to provide a clearer context.**
>
> **A1:** The two papers do not work on PLL, although their data can also be annotated with multiple labels. The similarity is that we all study datasets annotated with multiple labels. The differences between our work and theirs are:
> - **The problems studied are different.** First, the task of [1] is to predict soft labels given a dataset with crowdsourced soft labels. [2] works on noisy label learning, which aims to predict a single label given a dataset with label noise, and they are mainly single-label, while some can filter a single noisy label from multiple noisy labels. Our task (PLL) is to predict a single label given a dataset where each example has multiple candidate labels, among which only one label is correct. Therefore, the problem settings are different and we are the first benchmark for PLL.
>
> - **The evaluation goal is different.** [1] aims to evaluate a labeling model, which is a model trained to annotate labels. [2] mainly works on noisy label detection, which aims to find noisy label examples from the dataset. Our paper works on algorithm evaluation, which aims to evaluate the classification performance of different algorithms.
>
> - Our work also provides new model selection criteria with theoretical guarantees, which fills a gap in the PLL literature.
>
> Therefore, there are clear differences between these two papers and ours. We have cited them in the revised version of the paper.
>
> **Q2: Could the authors provide experimental evidence showing that the proposed criteria lead to better performance on the test dataset?**
>
> **A2:** The advantage is twofold. First, for the hyperparameter selection, we found that the test performance for our selected ones is better than the default hyperparameter in most cases. This is because we also add the default hyperparameter configuration to the selection pool, but always find that other hyperparameter settings are better than it. Second, early stopping helps us get better results. For example, from Tables 1 to 4, we can see that OA & ES is always better than OA. This conclusion is also true for CR and AA. Therefore, we can use our proposed criteria for early stopping to achieve better performance.
>
> **Q3: Would it be possible for the authors to include a numerical comparison showing how often each criterion achieves the best performance?**
>
> **A3:** In Table 1, CR wins 13 times, AA wins 3 times and OA wins 10 times. Similar results can also be obtained from other tables. We can see that CR can achieve the best performance in most cases. However, for specific algorithms and cases, we should specifically determine the best criterion, and the three criteria can be complementary to achieve better classification results.
>
> **Q4: Could the authors provide further explanation of $p(\boldsymbol{x},y)$ and $p(S|\boldsymbol{x}, y)$?**
>
> **A4:** $p(\boldsymbol{x},y)$ denotes the joint distribution of the instance $\boldsymbol{x}$ and the label $y$. $p(S|\boldsymbol{x}, y)$ denotes the conditional probability distribution of $S$ given $(\boldsymbol{x},y)$. We have added descriptions after Eq. (4) and the calculation of $p(S|\boldsymbol{x}, y)$ can be found in Equations (1) and (2) for example as well.
>
> **Q5: I want to bring two particular papers [3,4] into the author's sight.**
>
> **A5:** Thanks for the helpful reference. Both papers work on similar problem settings and we have cited them in the revised version.

---

> ### Comment · Reviewer_GD1c · 2024-11-22
> **Thanks for the reply**
>
> Thanks for the response. I have a following up question regarding Q1. I originally thought [1,2] to be similar to this paper, and the authors clarified it. After reading the response, I wonder if [3] is actually more similar. Essentially, I want to understand how PLL differs from the previous multi-label learning. Thanks.

---

> > ### Author Response · Authors · 2024-11-22
> > **Thanks for your question!**
> >
> > Thank you very much for your question. Considering multi-label learning [5] you mentioned, there are three problem settings where each training example is annotated with multiple labels.
> >
> > - **Partial-label learning (PLL)**: For each example, its multiple labels are different, and it is assumed that only one label is the true label, but not accessible to the learning algorithm. It is a multi-class classification problem, and the test data has only one label.
> >
> > - **Learning with multiple annotator noisy labels [3]**: For each example, some of its multiple labels may be the same because they are given by different people. It is a generalized form of noisy label learning, and each label can be true or false. It is also a multi-class classification problem, and the test data has only one label.
> >
> > - **Multi-label learning [5]**: For each example, its multiple labels are different and are all true labels. Unlike the previous two settings, the test data has multiple true labels, and our task is to predict multiple true labels for the test data.
> >
> > We hope this answer addresses your question. If you have any further questions or concerns, please do not hesitate to contact us. Thank you again for your help and time in reviewing our submission!
> >
> > Reference:
> >
> > [5] Zhang M L, Zhou Z H. A review on multi-label learning algorithms[J]. IEEE transactions on knowledge and data engineering, 2013, 26(8): 1819-1837.

---

> ### Author Response · Authors · 2024-11-26
> **Thanks again for reviewing our paper!**
>
> Dear Reviewer GD1c,
>
> We are very grateful that you have kindly raised your score on our submission. We sincerely appreciate the time and effort you have invested in reviewing our submission. Your comments and suggestions help us improve the quality of our work. Thank you again for your time and help!
>
> Best regards,
>
> ICLR 2025 Conference Submission5419 Authors

---

### Official Review · Reviewer_v9rL · 2024-10-29

**Soundness:** 3
**Presentation:** 3
**Contribution:** 3
**Rating:** 6
**Confidence:** 3

**Summary:**

The paper investigates the inconsistent evaluation of different partial-label learning methods and proposes a new benchmark for a fair evaluation. In particular, the standard setting in PLL does not have a validation set that consists of ground truth labels, while many recent studies employ such a clean validation set to tune their hyper-parameters. The paper, therefore, proposes "surrogate" accuracy metrics to replace the conventional prediction accuracy (because the conventional accuracy is intractable in training) in the validation to fine-tune hyper-parameters. Those alternative metrics are theoretically proved to be "close" to the exact one. Empirical evaluation shows that the many prior methods in PLL can achieve high performance or even out-perform most recent methods in the same benchmark.

**Strengths:**

The paper has pointed out a critical issue in PLL research where many recent studies try to achieve the state-of-the-art results by considering an unfair setting when benchmarking with other prior methods. This potentially misleads the research direction where the performance of those prior methods can even out-perform many recent PLL approaches.

In the standard setting of PLL, the ground truth labels are not available in validation sets. Hence, it is difficult to tune the hyper-parameters of the model of interest. Hence, the paper proposes some alternative metrics as a drop-in replacement of the standard accuracy to optimise the hyper-paramters of interest. Empirical evaluation shows that this approach results in high-performance models in many benchmarks. Note that this is a strong contribution of the paper because the proposed alternative metrics are proved to be "close" to the standard prediction accuracy under certain conditions.

**Weaknesses:**

**Confusing terminologies**: *model selection* vs *hyper-parameter tuning*

In the paper, the authors argue that the mismatch of the validation setting results in bad model selection. In fact, to what I understand, what that means is hyper-parameter tuning, not model selection. Hyper-parameter tuning is indeed a subset of model selection, but not the other way around. Model selection is a terminology in machine learning and also means things like variable selection and actual model choice (functional form, for example, should it be ResNet or DenseNet or a transformer), whereas the one discussed in the paper is to find the best model within a family. Thus, I suggest the authors to use the correct terminologies to reduce the confusion.

**Confusing toy example in Figures 1a and 1b**

The bar colors are inconsistent with the legend, making it hard to understand. In addition, why shouldn't the x-axis include the name of each method, instead of letting them floating inside the plotting area. The pilot (or toy) experiment mentioned at line 75 with the current description is hard to understand and should provide further details. For example, what are training and validation in Figures 1a and 1b mean? What are the meaning of the subcaptions in Figure 1? Are they the name of a method to obtain labels or the name of a dataset? I understand that the purpose is to demonstrate that evaluating methods on two different settings: with and without clean validation set leads to different performance. However, the explanation is non-coherent and hard to understand.

**Complicating notations**

Eq. (5): accuracy is already an expected (or average) value itself as shown in the right hand side of Eq. (5). Thus, the notation: $\mathbb{E}[\mathrm{ACC}(f)]$ seems cumbersome, and can be simplified to ACC only.

**Proof of Proposition 1:**

The second equality is unclear. Why is it possible to make the subtraction of two expectations be a single expectation? This is quite problematic because according to Lemma 1, S is dependent on x and y.

Another issue related to Proposition 1 is Lemma 1. A lemma is introduced as a stepping stone to use its **conclusion** to prove meaningful results (e.g., a theorem). Although Lemma 1 is introduced to prove Proposition 1 at line 192, I do not see any connection to the proof of the Proposition 1. In particular, the proof just uses the assumption (or the definition to be exact) of Lemma 1 about the *ambiuguity degree* to include in the result of Proposition 1. Hence, stating Lemma 1 to lead to the result in Proposition is misleading.

**Unclear distinction between two synthetic datasets**
Lines 286 and 287: It is unclear about the differences between the two approaches: aggregate and vaguest. In the text, they are almost equivalent or identical. Could the authors provide detailed clarification about their differences?

**Questions:**

As explained in the **Weaknesses**, could the authors clarify the following concerns:
- Provide a comprehensive and coherent description of the toy example in Figures 1a and 1b (or the one at line 75)
- Clarify further the proof of the Proposition 1 because it seems to be the main building block for the subsequent theoretical results.
- Clarify the differences between the two synthetic datasets at line 286.

---

> ### Author Response · Authors · 2024-11-22
> **Response To Reviewer v9rL**
>
> First of all, we would like to express our sincere gratitude for your efforts and time in reviewing our submission. Below are the responses to your questions and comments.
>
> **Q1: Confusing terminologies: model selection vs hyper-parameter tuning.**
>
> **A1:** We agree that it is helpful to express more carefully in this paper that the proposed criteria are mainly for hyperparameter selection. We have added it in the paper. Besides, our wording mainly refers to DomainBed [1], which also proposes some model selection criteria for hyperparameter tuning. Although we mainly use them to select hyperparameters, we can also use them to select appropriate models from a model pool. For example, we can determine which of ResNet and DenseNet is better based on our criteria, since the deeper one (DenseNet) may not always be a better choice according to our experimental results. We believe that our proposed criteria can be used in broader aspects than just hyperparameter selection.
>
> **Q2: Confusing toy example in Figures 1a and 1b.**
>
> **A2:** We apologize for our unclear presentation. The lighter and darker colors indicate different ways to use a clean validation set for a given algorithm. The lighter colors indicate that we use it only to tune the hyperparameters, and the darker colors indicate that we add it to the training set without carefully tuning the hyperparameters. We can see that using it for training can often lead to better results for many PLL algorithms, showing that current experimental settings of PLL may not be appropriate. The floating text means used algorithms, we float them to align them to the two right panels since the lengths of the algorithm names are very long and can take up a lot of space. We have revised the captions and figures to make them clearer.
>
> **Q3: Complicating notations.**
>
> **A3:** We agree with you and have revised $\mathbb{E}\left[{\rm ACC}(f)\right]$ to ${\rm ACC}(f)$.
>
> **Q4: Proof of Proposition 1.**
>
> **A4:** First, we have revised Lemma 1 to Definition 2 as you suggested. Then we explain the proof of Proposition 1 in Appendix A.1. The key idea of it is to unfold the difference as an expectation for a term for each data point and then upper bound the values by our introduced definitions. The first equation is according to the definitions there. The second is natural if the wrong prediction is in the candidate label set. The third equation can be obtained by traversing as said in the paper. The fourth equation is by the definition of expectations. The fifth equation is by the conditional probability equation, and the last inequality is by the definition of Ambiguity Degree.
>
> **Q5: Unclear distinction between two synthetic datasets.**
>
> **A5:** Sorry for the error. We have corrected the sentence. The correct phrase should be: The second is PLCIFAR10-Vaguest, which assigns each example the largest candidate label set from the annotators. The two datasets are different.
>
> Reference:
>
> [1] In Search of Lost Domain Generalization, ICLR 2021.

---

> > ### Comment · Reviewer_v9rL · 2024-11-25
> > **Acknowledge the clarification**
> >
> > Thank you, the authors, for the clarification. That has addressed most of my concerns. Although I am not very familiar to the topic, the paper is quite positive. I keep my rating as it.

---

> > > ### Author Response · Authors · 2024-11-26
> > > **Thanks again for reviewing our paper!**
> > >
> > > Dear Reviewer v9rL,
> > >
> > > We really appreciate the time and effort you've put into evaluating our submission. Your thorough response is extremely helpful for our work. Thank you again for your invaluable help and time!
> > >
> > > Best regards,
> > >
> > > ICLR 2025 Conference Submission5419 Authors

---

### Official Review · Reviewer_RWfx · 2024-10-31

**Soundness:** 3
**Presentation:** 3
**Contribution:** 3
**Rating:** 8
**Confidence:** 3

**Summary:**

This paper introduces PLENCH, the first Partial-Label learning bENCHmark for comparing
state-of-the-art deep PLL algorithms.

Authors create Partial-Label CIFAR-10 (PLCIFAR10), an image dataset of human-annotated partial labels collected from Amazon Mechanical Turk. Furthermore, paper investigates the model selection problem for PLL, and proposes model selection criteria with theoretical guarantees.

Key takeaways indicate that simpler algorithms can sometimes match or exceed complex ones, no single algorithm excels in all scenarios, and model selection practices are crucial for fair comparisons in PLL studies.

**Strengths:**

- The authors make a significant contribution by being the first to systematically investigate model selection problems in partial-label learning (PLL), addressing a gap in the existing literature. The paper presents a comprehensive PLL benchmark that includes 27 algorithms and 11 real-world datasets.
- A notable strength of the paper is the introduction of PLCIFAR10, a new benchmark dataset for PLL featuring human-annotated partial labels. This dataset provides an effective and realistic testbed for evaluating the performance of PLL algorithms in scenarios that closely mimic real-world conditions.

**Weaknesses:**

N/A

**Questions:**

What changes do you expect to see on larger datasets and with more complex deep neural networks?

---

> ### Author Response · Authors · 2024-11-22
> **Response To Reviewer RWfx**
>
> First of all, we greatly appreciate your time and efforts in reviewing our paper. We am encouraged and thankful that you agreed with the contributions of the paper. Below are the responses to your questions and comments.
>
> **Q1: What changes do you expect to see on larger datasets and with more complex deep neural networks?**
>
> **A1:** Thank you for your valuable question. Using larger datasets with more complex patterns is very promising for partial-label learning (PLL). However, the performance of many PLL methods may decrease because they have made simple assumptions about the data generation process that may not be compatible with large complex datasets. Therefore, it is promising to evaluate PLL methods with larger datasets in more realistic scenarios. We will consider this as our future work.
>
> The use of more complex deep neural networks is also a promising direction. In this paper, we mainly follow previous work to consider ResNet and DenseNet. Using deeper and larger networks can improve the classification performance, and we will also consider them in the future. Moreover, although the use of larger networks is promising, it may also lead to a higher risk of overfitting problems. Then it is helpful to use our proposed model selection criteria for early stopping.

---

> > ### Comment · Reviewer_RWfx · 2024-11-26
> > **Acknowledge the clarification**
> >
> > Thank you for the clarification. All of my concerns are resolved.

---

> > > ### Author Response · Authors · 2024-11-26
> > > **Thanks again for reviewing our paper!**
> > >
> > > Dear Reviewer RWfx,
> > >
> > > We really appreciate the time and effort you've taken to review our submission. Your insightful feedback helps us to improve our work. Thank you again for your help and time!
> > >
> > > Best regards,
> > >
> > > ICLR 2025 Conference Submission5419 Authors

---

### Official Review · Reviewer_oKAk · 2024-11-01

**Soundness:** 3
**Presentation:** 4
**Contribution:** 4
**Rating:** 8
**Confidence:** 4

**Summary:**

The paper proposes the partial label learning (PLL) benchmark PLENCH that standardizes experiments of PLL. The related work in PLL mostly selects hyperparameters based on a normally labeled validation dataset, which is highly unrealistic. The authors  propose three model selection criteria that are theoretically analyzed and empirically investigated. Furthermore, the authors propose a new dataset PLCIFAR10 that consists of human-annotated partial labels. It serves for a more realistic evaluation, as  partial labels are not synthetically generated. In their experiments, they include an excessive amount of PLL algorithms across several datasets.

**Strengths:**

- The paper is well-written and accessible.
- Model selection in PLL is a relevant, underexplored topic.
- The proposed dataset in a realistic PLL setting adds significant value to the PLL literature.
- Despite the model selection criteria's simplicity, the authors establish a solid theoretical link to expected accuracy and demonstrate its benefits.
- The experiments are extensive, covering many PLL algorithms from top venues.

**Weaknesses:**

While I think that this paper is really good, it would be nice if the authors can clarify the following weaknesses:
- **Model selection criteria:** While the criteria are intuitive and theoretically analyzed, presenting Oracle Accuracy as a contribution is problematic. As also noted by the authors, OA is standard in the literature and, thus, it should be presented as a baseline, not as a novel contribution. IMO the contribution statement in the introduction needs to be slightly adapted.
- **Use of early stopping:** For me, it is unclear how ES is used in the experiments. Lines 245-247 suggest it is applied to everything except OA. So, my question is: Is ES used in the case of CR and AA? Maybe the authors can clarify this a little better in the paper?
- **Difference between Aggregate and Vaguest not clear:** I can not find a difference in the explanation between the Aggregate and the Vaguest version. The authors state:

    > The first is PLCIFAR10-Aggregate, which assigns **the aggregation of all partial labels from all annotators to each example**. The second is PLCIFAR10-Vaguest, which assigns **to each example the aggregation of all partial labels from all annotators.**
    >

    Considering this explanation, there should be no difference between the versions. It would be nice if the authors could clarify (or fix) this.

- **Minor Stuff:**
    - The subfloats in Figure 2 are not aligned at the top.

**Questions:**

**Questions**
- Can you provide feedback to the weaknesses I mentioned above?
- I was wondering if there is access to the identities of the annotators and how the authors ensured their privacy?

**Details Of Ethics Concerns:**

Is it correct that crowd workers received only $0.01 for labeling 10 images? If so, this rate seems to be considerably lower than comparable annotation campaigns on noisy label learning:
- In [1], workers received $0.08 for labeling 10 images, which is 8 times higher.
- In [2], the payment was $0.03 for 10 images, also 3 times higher.
- In [3], the annotators receive hourly payment of $8. Considering the payment in this submission, an annotator has to provide about 2.2 labels per second to match this hourly rate.

[1] Peterson, Joshua C., et al. "Human uncertainty makes classification more robust." Proceedings of the IEEE/CVF international conference on computer vision. 2019.
[2] Wei, Jiaheng, et al. "Learning with noisy labels revisited: A study using real-world human annotations." arXiv preprint arXiv:2110.12088 (2021).
[3] Collins, Katherine M., Umang Bhatt, and Adrian Weller. "Eliciting and learning with soft labels from every annotator." Proceedings of the AAAI conference on human computation and crowdsourcing. Vol. 10. 2022.

Can you please provide rational on how you selected the payment rate?

---

> ### Author Response · Authors · 2024-11-22
> **Response To Reviewer oKAk**
>
> First of all, we would like to thank you for your time and effort in reviewing our paper. We are very grateful for your helpful and high quality comments and suggestions. Below are the answers to your questions and comments.
>
> **Q1:OA should be presented as a baseline, not as a novel contribution.**
>
> **A1:** We agree with your comment. We have modified the paper to show that OA is not contribution.
>
> **Q2: It is unclear how ES is used in the experiments. Is ES used in the case of CR and AA?**
>
> **A2:** Following DomainBed [1], we used ES for CR and AA. The detailed descriptions can be found in Section 5.3. Specifically, we recorded the values of CR and AA on a validation set and the test accuracy for all iterations. We then selected the iteration with the highest value of a criterion and returned the corresponding test accuracy. Notably, the selected iteration may be in the middle stage of training, which can have similar effects to ES.
>
> In real applications, we can also check the values of these criteria and perform ES when we find that the value of these criteria starts to decrease.
>
> **Q3: Difference between Aggregate and Vaguest not clear.**
>
> **A3:** Sorry for the error. We have corrected the sentence. The correct phrase should be: The second is PLCIFAR10-Vaguest, which assigns each example the largest candidate label set from the annotators.
>
> **Q4: The subfloats in Figure 2 are not aligned at the top.**
>
> **A4:** We have revised the figure as you suggested.
>
> **Q5: I was wondering if there is access to the identities of the annotators and how the authors ensured their privacy?**
>
> **A5:** Annotators' identities are not accessible through the platform interface. Their privacy is strictly protected by the Amazon MTurk platform.
>
> **Q6: Can you please provide rational on how you selected the payment rate?**
>
> **A6:** The payment rate was chosen based on our funding and the characteristics of the problem setting. We first experimented with this payment rate on a small amount of data and found that the noise rate was similar to previous work using different payment rates [2]. For example, on the CIFAR-10 dataset, CIFAR-10N-random has a noise rate of 18%, while our PLCIFAR10-Vaguest dataset has a noise rate of 17.56%. Therefore, we thought that the annotation quality of our payment rate might be comparable to previous work. In addition, we found that the tasks were completed quickly in a short period of time, so we ultimately set this payment rate. Thank you for providing the references and we have cited them in the paper.
>
> Reference:
>
> [1] In Search of Lost Domain Generalization, ICLR 2021.
>
> [2] Learning with Noisy Labels Revisited: A Study Using Real-World Human Annotations, ICLR 2022.

---

> ### Comment · Reviewer_oKAk · 2024-11-25
>
> Thanks for your detailed and thoughtful responses. The explanations have effectively addressed all the raised points and resolved my concerns.

---

> > ### Author Response · Authors · 2024-11-26
> > **Thanks again for reviewing our paper!**
> >
> > Dear Reviewer oKAk,
> >
> > We are deeply grateful for the effort and time you've devoted to evaluating our submission. Your insightful feedback contributes significantly to enhancing our work. Thank you once more for your help and time!
> >
> > Best regards,
> >
> > ICLR 2025 Conference Submission5419 Authors

---

### Author Response · Authors · 2024-11-22
**General Response**

First of all, we would like to thank all the area chairs and reviewers for their great efforts and time in reviewing our paper. Our paper has benefited greatly from their insightful and valuable suggestions. We have revised the manuscript according to the reviewers' suggestions and marked them in blue in the new version of the manuscript. The main differences include

- We have corrected our wording about PLCIFAR10-Vaguest:

> The second is PLCIFAR10-Vaguest, which assigns each example the largest candidate label set from the annotators.

- We revised $\mathbb{E}\left[{\rm ACC}(f)\right]$ to ${\rm ACC}(f)$ for simplicity.

- Revised Lemma 1 to Definition 2 for better logic.

- We revised and added clearer descriptions of Figure 1 and modified Figure 2 for better alignment.

- We expressed that model selection criteria are mainly used for hyperparameter tuning in this paper.

- We excluded OA as our contribution.

- We added several related references.

Thank all the reviewers all again for their help and suggestions on this submission!

---

### Meta-Review · Area_Chair_YWZi · 2024-12-18

**Metareview:**

This work addresses key challenges in Partial-Label Learning (PLL) by introducing PLENCH, the first comprehensive benchmark for comparing deep PLL algorithms. It proposes novel model selection criteria with theoretical guarantees, standardizes evaluation settings, and introduces PLCIFAR10, a real-world image dataset with human-annotated partial labels, enabling fair and practical assessment of PLL methods.

All the reviewers agree this is a good paper. I think it should be accepted to ICLR and I would encourage the authors to clearly address all the issues raised by the reviewers in the camera ready.

**Additional Comments On Reviewer Discussion:**

All the reviewers agree this is a good paper. I think it should be accepted to ICLR and I would encourage the authors to clearly address all the issues raised by the reviewers in the camera ready.

---

### Decision · Program_Chairs · 2025-01-22

Accept (Spotlight)